## Comparison of calibration methods of a PICO basal ice shelf melt module implemented in the GRISLI v2.0 ice sheet model

Maxence Menthon<sup>1</sup>, Pepijn Bakker<sup>1</sup>, Aurélien Quiquet<sup>2</sup>, Didier M. Roche<sup>1, 2</sup>, and Ronja Reese<sup>3</sup>

**Correspondence:** Maxence Menthon (max.menthon@gmail.com)

Abstract. Uncertainties in future sea level rise are mainly due to uncertainties in Antarctic ice sheet projections. Indeed, modelling the future of the Antarctic ice sheet presents many challenges. One of them is being able to model the physical interactions between the ocean and the ice shelves. As a result of technical challenges related to computational resources, implementation and different modelling time-scales, these interactions are often parameterised rather than explicit resolved in ice sheet models. These parameterisations vary in complexity and calibration method, eventually leading to differences in resulting sea level rise contribution of several meters. Here we present the implementation of the PICO basal ice shelf melt module in the GRISLI v2.0 ice sheet model. We compare six different statistical methods to calibrate PICO and assess how robust these methods are if applied at different resolutions and areas of the Antarctic ice sheet. We show that computing the Mean Absolute Error of the bins is the best method as it allows us to match the entire distribution of melt rates retrieved from satellite data at different resolutions as well as for different Antarctic ice shelves. It also results in a smaller parameter space than the other tested methods. This method makes use of melt rate bins and minimizes the differences between the values of the bins of the model and the ones of the observational target. It gives equal weight to the full distribution of melt values, low, medium and high values. We find that, using this method, region-specific calibration of ice-ocean interactions is not needed and we can avoid using ocean temperature bias corrections. Finally, we assess the impact of the implementation of PICO in GRISLI and of the calibration choice on future projections of the Antarctic ice sheet up to the year 2300.

Copyright statement. TEXT

#### 1 Introduction

The future evolution of the Antarctic ice sheet is the largest uncertainty in sea level rise projections for the end of the century (Edwards et al., 2021). The mass loss of the Antarctic ice sheet is primarily driven by basal melting of ice shelves (Pritchard et al., 2012). The ice shelves have a buttressing effect, slowing the ice flow towards the ocean (Dupont and Alley, 2005). Their thinning observed over the last decades (Rignot et al., 2013; Paolo et al., 2015; Adusumilli et al., 2020) is due to increased

<sup>&</sup>lt;sup>1</sup>Department of Earth Sciences, Faculty of Science, Vrije Universiteit Amsterdam, Amsterdam, The Netherlands

<sup>&</sup>lt;sup>2</sup>Laboratoire des Sciences du Climat et de l'Environnement, LSCE/IPSL, CEA-CNRS-UVSQ, Université Paris-Saclay, 91191 Gif-sur-Yvette, France

<sup>&</sup>lt;sup>3</sup>Department of Geography and Environmental Sciences, Northumbria University, Newcastle, UK

heat provided by circumpolar deep water in the cavities directly beneath the ice shelves (Schmidtko et al., 2014; Stewart and Thompson, 2015; Jenkins et al., 2018). Sub-surface melt of the ice shelves impacts the oceanic circulation in the ice cavities as well as larger scale oceanic circulation (Bennetts et al., 2024). These feedbacks and the large range of spatio-temporal scales at play, from turbulence to large-scale ocean circulation, make the ice-ocean interaction a complex process challenging to model accurately (Bennetts et al., 2024). Additionally, on a retrograde bathymetry, such as in West Antarctica, the thinning of ice shelves and retreat of the grounding line can trigger marine ice sheet instabilities (Weertman, 1974; Schoof, 2007) leading to irreversible commitment to sea level rise. Hence, understanding and having the ability to model the ice-ocean interaction accurately is crucial to constrain uncertainties of projections of the future rise in sea level.

With our current computational resources, it is necessary to use parameterisations in ice sheet models to compute the physical interactions between the ocean and the ice. Over the last decade, several basal melt parameterisations have been developed and implemented in ice sheet models with different complexities in the simplification of the ocean circulation beneath the ice-shelves and its physical interaction with the ice (Reese et al., 2018a; Lazeroms et al., 2019; Pelle et al., 2019; Jourdain et al., 2020; Lambert et al., 2023). Berends et al. (2023) demonstrated that the choice of the sub-shelf melt parameterisation has a strong impact on the Antarctic ice sheet retreat for idealised as well as realistic geometries. Moreover, the values of the parameters are poorly constrained, and in some cases the parameterisations require ocean temperature corrections up to 2 K to be able to match the basal melt rates observed (Jourdain et al., 2020; Reese et al., 2023).

Here, we present the implementation of the Postdam Ice-shelf Cavity mOdel (PICO) (Reese et al., 2018a) in the GRISLI v2.0 ice sheet model (Quiquet et al., 2018) but also a comparison of methodologies to calibrate PICO. We aim at calibrating the module to match the whole distribution of values from the observations to the extent possible. In this research no ocean temperature corrections are added in the calibration process in order to have a more physical relationship between the forcing and the computed melt rates. The article presents first the methodology in section 2, then the results in detail in section 3 including: comparison of calibrations methods, sensitivity estimates and future projections. Section 4 discusses the limitations and possible improvements for next studies. Finally, section 5 concludes and gives some perspectives.

#### 45 2 Methodology

30

#### 2.1 PICO basal ice shelf melt module

PICO is a parameterisation that computes the basal melt rates under the ice shelves. It is described in details in Reese et al. (2018a). We present here only the key concepts of PICO. It is a box model, based on the work of Olbers and Hellmer (2010). The ice shelves are divided into boxes, and the shape and number of boxes in one ice shelf dependend on two variables: distance to the ice shelf front and distance to the grounding line. The number of boxes  $n_D$  in one ice shelf D is then defined by:

$$n_D = 1 + \operatorname{rd}\left(\sqrt{d_{GL}(D)/d_{\max}}(n_{\max} - 1)\right)$$
(1)

with  $d_{\rm GL}$  the distance of each grid cell to the grounding line,  $d_{\rm max}$  the maximum distance between the grounding line and the ice shelf front among all the ice shelves of the ice sheet, and  $n_{\rm max}$  the maximum number of boxes kept here at 5.

PICO accounts for one-dimensional overturning circulation in ice-shelf cavities (Lewis and Perkin, 1986). The overturning flux under the ice shelf q is driven by the density difference between the ocean box B0 ( $\rho_0$ ) and the first box B1 under the ice shelf at the level of the grounding line ( $\rho_1$ ):

$$q = C(\rho_0 - \rho_1) \tag{2}$$

The value of q must be greater than zero. A single overturning flux value is calculated for all boxes of the same ice shelf. The constant overturning coefficient C (Sv m³ kg $^{-1}$ ) captures effects due to friction, rotation and bottom form stress, more details are given in Olbers and Hellmer (2010). C is one of the two PICO parameters that we calibrate in the present study. To compute basal melt rates  $m_k$  in the box  $B_k$ , PICO requires 2 ocean inputs: ocean temperature  $T_{k-1}$  and salinity  $S_{k-1}$ ; and one ice sheet input: the ice draft to calculate the pressure at the ice-ocean interface under the ice shelf  $p_k$  using  $p_k = \rho_{SeaWater} \cdot g \cdot z_{IceDraft}$ , with  $\rho_{SeaWater} = 1033$  kg m $^{-3}$ . For the box  $B_1$ , the ocean inputs are the average temperature ( $T_0$ ) and salinity ( $S_0$ ) at the continental shelf depth in front of the corresponding ice shelf. For the next boxes  $B_k$ , the forcing temperature and salinity depend on the overturning q and the temperature and salinity computed for the previous box  $T_{k-1}$  and  $S_{k-1}$ . The details of the analytical derivation are given in Reese et al. (2018a) in appendices A and B. The melt rate in the box k is then computed as follows:

$$m_k(x,y) = -\frac{\gamma_T^*}{\nu^{\lambda}} \left( aS_{k-1} + b - cp_k(x,y) - T_{k-1} \right)$$
(3)

where  $\gamma_{\rm T}^*$  is the heat exchange coefficient (m.s<sup>-1</sup>), the second of the two PICO parameters calibrated in this paper,  $\nu = \rho_i/\rho_w \sim 0.89$ ,  $\lambda = L/c_p \sim 84$  °C. The coefficients from linearisation of the equation of state for the freezing point of seawater are: a is the liquidus slope coefficient and equals -0.0572 °C PSU<sup>-1</sup>, b is the liquidus intercept coefficient and equals 0.0788 °C, c is the liquidus pressure coefficient and equals  $7.77 \times 10^{-8}$  °C Pa<sup>-1</sup>.

#### 2.2 Choices for the implementation of PICO in GRISLI

To implement PICO in GRISLI and to select the values of the two parameters (C and  $\gamma_T^*$ ) we make some choices that differ from Reese et al. (2018a). We give an overview of these choices here. All the PICO implementation has been done in Fortran90 to corresponds to GRISLI development language.

First, the heaviest computation part in the module is the computation of the geometry of the ice shelves. To reduce computational costs we avoid identifying individual ice shelves at each time step, instead we decide to solve one instance of the PICO equations per drainage basin, with the basins as presented in Mouginot and Rignot (2017) and used in Rignot et al. (2019), with the difference that we combine the two drainage basins of the two largest ice shelves as done by Jourdain et al. (2020): Ronne with Filchner ice shelves and Ross East with Ross West ice shelves. Each drainage basin defines the external borders of the geometry of the ice shelves and is used to compute the oceanic forcing inputs. This implies that if two different ice shelves are in a same drainage basin, they are seen as one ice shelf for PICO. Inversely, if one ice shelf has two drainage basins, it is then seen as two separate ice shelves for PICO. The number of boxes in each drainage remains relative to the maximum distance between the ice shelf front and the grounding line of all the ice shelves, as defined in the equation 1. And since within the same

drainage basin there are roughly similar sizes of ice shelves, the division by drainage basin do not cause discrepancies such as small ice shelves with up to five boxes or larger ice shelves with few boxes (see Supplement Figure S1). In all cases, the ice shelf front and the grounding line are defined in the same way by being neighbors to open ocean or grounded grid cells, respectively. The geometry of the ice shelves is recomputed at every time steps to readjust the boxes to the changing grounding line and ice shelf front position. This simplification of solving one PICO instance per drainage basin enables us to compute faster for each ice shelves their number of boxes, as well as their corresponding temperature and salinity inputs.

Then, Reese et al. (2018a) used four selection criteria to calibrate PICO and define the values of the PICO parameters C and  $\gamma_{\rm T}^*$ . The two first criteria are: (1) to not have freezing dominating the melt rates values in the first ice shelf box and (2) the overall mean basal melt rates must decrease between the first and second box of the ice shelf. The criteria (3) and (4) are constraints on the average values the melt rates that should be in the cold ice cavities of Filchner-Ronne Ice Shelf (FRIS) and in the warm ice cavities of Pine Island glacier respectively. Here, we do not follow any of these criteria, we apply different criteria. The rational is that while the PICO equations assume that (1) and (2) are true for the one horizontal dimensional case, melt rate patterns are complex in two horizontal dimensions. Indeed, the retrieved basal melt rates from remote sensing (Adusumilli et al., 2020; Paolo et al., 2023) show refreezing in some areas close to the grounding line but also higher melt rates close to the ice-shelf front. Instead of criteria (3) and (4), the calibration methods we tested here (presented in section 2.5) are designed to be able to capture the whole distribution of values, not only the average values, at an Antarctic wide scale as well as at an ice shelf scale.

The grounding line can be defined in different ways and therefore can lead to different PICO boxes geometries. Here, in GRISLI-PICO, we consider any ice points to be on the grounding line if it has some neighbors that are grounded and others that are floating, and as ice front any ice point that is floating and adjacent to ocean. In contrast, in PISM-PICO Reese et al. (2018a) did not include the grounding line of ice rises and also excluded holes in ice shelves as ice-shelf front when identifying PICO boxes. The grounding lines of ice rises are defined as not being directly connected to the main grounded part of the ice sheet which is identified by the size of the connected grounded region. Thus, in PISM-PICO it is possible to have ice shelves without grounding line connected to the main ice sheet, where PICO cannot define a box geometry. In these places, the parametrisation of Beckmann and Goosse (2002) was used in PISM to have a rough estimate of the basal melt rates.

GRISLI incorporates a dynamic calving front that advances based on a balance between the Lagrangian ice flux and local surface and basal mass balances (Quiquet et al., 2018). To evaluate potential ice shelf advance at each timestep, the model must compute these mass balances even beyond the current ice extent. Unlike alternative approaches such as the level set method, which do not require mass balance information outside the ice mask, this is a necessary feature for GRISLI. Thus, in regions beyond the ice shelf, over open ocean, we apply the parameterisation of DeConto and Pollard (2016), defined as follows:

$$m = \frac{K_T \rho_w C_w}{\rho_i L_f} |T_o - T_f| (T_o - T_f) \tag{4}$$

115

The main difference is the inclusion of a quadratic dependence between the melt rate and the difference of the temperature between the ocean  $T_o$  and the ocean freezing point at the ice base  $T_f$ . This quadratic relation enables us to limit the growth of the ice shelves towards the ocean. The combined factor  $\frac{K_T \rho_w C_w}{\rho_i L_f}$  equals to 0.224 m yr<sup>-1</sup> °C<sup>-2</sup> as we keep the same  $K_T$ 

value of 15.77 m yr<sup>-1</sup> °C<sup>-1</sup> from DeConto and Pollard (2016) and Pollard and Deconto (2012). The temperature input  $T_0$  is computed the same way as for PICO. The parameterisation chosen for the open ocean does not impact the calibration results but does impact the transient ice sheet simulations.

Finally, in GRISLI v2.0 the iceberg calving is defined by a simple ice thickness threshold criterion (Quiquet et al., 2018).

The threshold value varies in space and time as it is dependent on the depth of the bathymetry at the location of the ice shelf front.

#### 2.3 Calibration ensemble


To calibrate the two PICO parameters C and  $\gamma_{\rm T}^*$  we run an ensemble of 169 members of PICO implemented in GRISLI corresponding to all possible combinations between 13 values for the parameter C (ranging from 0.01 Sv m³ kg $^{-1}$  to 15.00 Sv m³ kg $^{-1}$ ) and 13 values for the parameter  $\gamma_{\rm T}^*$  (from  $0.01 \times 10^{-5}$  m s $^{-1}$  to  $15.00 \times 10^{-5}$  m s $^{-1}$ ). The range of values for the two parameters has been chosen based on literature (Reese et al., 2018a; Burgard et al., 2022; Reese et al., 2023) and adjustments in such a way that the best values are not on one of the extremes of the range of values. The geometry of the ice sheet and the ice shelves is kept fixed to remove the influence of ice shelves geometry changes on the computed basal melt rate. The fixed geometry corresponds to Bedmap2 (Fretwell et al., 2013) with a 30-year relaxation with GRISLI. This relaxation is needed because with the basal drag coefficient inversion methodology used for ISMIP6, we compute the ice sheet internal thermal equilibrium with a long (60 kyr) experiment with fixed observed geometry. Thus, to avoid any artificial drift when releasing this constraint we run a 30 years relaxation experiment with the same boundary conditions as for the control experiment from ISMIP6 (Seroussi et al., 2024). Doing the calibration of PICO in a coupled GRISLI-PICO with fixed geometry enables to facilitate the transition between the PICO calibration and the GRISLI-PICO transient experiments (see section 2.6) without impacting the results. With this ensemble of 169 simulations, we apply six different methods, presented in subsection 2.5, to evaluate which members of the ensemble provide the best fit with respect to the observational dataset.

#### 2.4 Data: ocean forcing and basal melt rates target

The oceanic forcing we use for the calibration ensemble presented above in subsection 2.3 is the dataset produced by Jourdain et al. (2020). This is a present-day estimate of three-dimensional fields of temperature and salinity of the ocean surrounding the Antarctic ice sheet. Jourdain et al. (2020) computed this estimate by using the following data sets: a pre-release of NOAA World Ocean Atlas 2018 covering the period 1995-2017 (Locarnini et al., 2018; Zweng et al., 2019), the Met Office EN4 subsurface ocean profiles for the period 1995-2017 (Good et al., 2013), and Marine Mammals Exploring Oceans from Pole to Pole for the period 2004 to 2018 (Treasure et al., 2017). The final dataset created by Jourdain et al. (2020) includes extrapolation of the ocean properties into the ice shelf cavities where observations are not available. The end product is on a polar stereographic grid with a resolution of 8 km horizontally and 60 m vertically.

Our target is the average basal melt rates retrieved by Adusumilli et al. (2020), a dataset that used CryoSat-2 altimetry to create an average value estimate of basal melt rates of the ice shelves of Antarctica for the period 2010-2018 at a resolution of 500 m.

These ocean forcings and basal melt rates target differ from Reese et al. (2018a) where they used Schmidtko et al. (2014) for the ocean forcing and Rignot et al. (2013) as the target for the basal melt rates. The selected datasets in the present study are more up to date and have good overlap with the time period of data retrieval between the forcings and the target. For all the methods and analysis, all the datasets, the forcings and the observational target, are up-scaled to the same resolution as the ice sheet model (16 km or 40 km), using CDO bilinear interpolation.

### 2.5 Six statistical methods of calibration of the two PICO parameters: C and $\gamma_{ m T}^*$



The six statistical methods of calibration compared in this study are explained here. The overview of the methods is given in Table 1, including names, equations, and short description. We present first the three methods that do not use binning of melt rates, then how we process the binning, and finally the three methods that use binning. For all the methods, the model data and the observational data are expressed in m yr<sup>-1</sup> (m of ice equivalent per year). The analysis methods that do not use binning are the following:

- Absolute Difference of Averages (ADA): we compute the average value of each ensemble member and of the target, and compute the absolute difference between each ensemble member average with respect to the average of the target.
  - Two-dimensional Root Mean Square Error (2D RMSE): we compute the RMSE cell-to-cell with the same geographical location between each ensemble member and the target.
- Two-dimensional Mean Absolute Error (2D MAE): we compute the MAE cell-to-cell with the same geographical location between each ensemble member and the target. Since no squaring is used in the error computation of the MAE, the
   MAE is less sensitive to outliers than the RMSE.

| Methods name       | Statistical formulas                                                                                          | Description of methods to rank the ensemble members         |
|--------------------|---------------------------------------------------------------------------------------------------------------|-------------------------------------------------------------|
| ADA                | $\frac{1}{n} \sum_{i=1}^{n} x_{member,i} - \frac{1}{n} \sum_{i=1}^{n} x_{target,i}$                           | Lowest ADA (Absolute Difference of Averages) between        |
|                    |                                                                                                               | each ensemble member and the target                         |
| 2D RMSE            | $\sqrt{\frac{1}{n}\sum_{i=1}^{n}\left(x_{member,i}-x_{target,i}\right)^{2}}$                                  | Lowest value of the RMSE (Root Mean Square Error) be-       |
|                    | •                                                                                                             | tween each ensemble member grid cells and the target grid   |
|                    |                                                                                                               | cells, with 2D geographical correspondence                  |
| 2D MAE             | $\frac{1}{n} \sum_{i=1}^{n}  x_{member,i} - x_{target,i} $                                                    | Lowest value of MAE (Mean Absolute Error) between each      |
|                    |                                                                                                               | ensemble member grid cells and the target grid cells, with  |
|                    |                                                                                                               | 2D geographical correspondence                              |
| ADA of bins        | $\frac{1}{m} \sum_{j=1}^{m} \bar{B}_{\text{member},j} - \frac{1}{m} \sum_{j=1}^{m} \bar{B}_{\text{target},j}$ | Lowest ADA of bins (Absolute Difference of Averages of      |
|                    |                                                                                                               | the bins) between the bins of each ensemble members and     |
|                    |                                                                                                               | the bins of the target                                      |
| RMSE of bins       | $\sqrt{\frac{1}{m}\sum_{j=1}^{m} \left(\bar{B}_{\text{member},j} - \bar{B}_{\text{target},j}\right)^2}$       | Lowest value of the RMSE (Root Mean Square Error) be-       |
|                    | •                                                                                                             | tween the bins of each ensemble member and the bins of      |
|                    |                                                                                                               | the target                                                  |
| MAE of bins        | $\frac{1}{m}\sum_{j=1}^{m}\left \bar{B}_{\mathrm{member},j}-\bar{B}_{\mathrm{target},j}\right $               | Lowest value of the MAE (Mean Absolute Error) between       |
| T.11. 1 Co d' d' 1 |                                                                                                               | the bins of each ensemble member and the bins of the target |

**Table 1.** Statistical methods applied to rank the ensemble members compared to the target (observations): names, equations and short descriptions. Where n is the number of grid cells existing in both data sets,  $x_{member,i}$  (respectively  $x_{target,i}$ ) refers to any single grid cell in one ensemble member (grid cell in the target), m is the number of bins (10 in this study),  $\bar{B}_{member,j}$  is each bin of a single ensemble member. The same nomenclature is applied for ensemble members and the observations.

By applying the three first ranking methods, the ranking metrics do not enable to pick systematically the ensemble members with the best fit to the distribution of values of the observational dataset. To improve that, we decide to bin each datasets: the ensemble members as well as the target. The aim of the binning is to be able to force the method to pick ensemble members that fit better the target distribution, including the higher and lower tails of the distribution. We proceed with the binning of the melt rates as shown in the schematic Figure 1. Each bin  $(\bar{B})$  is the average value of 10% of the total number of ordered data points. The data points must be ordered to make each bin representative of a specific share of the dataset. For instance, if an ensemble member has 200 data points, the 20 points with the lowest melt-rate values are averaged and become one bin value. We proceed similarly for all the following bins and end up with 10 bins for each of the 169 members of the ensemble  $\bar{B}_{\text{member},j}$  and 10 bins for the target too  $\bar{B}_{\text{target},j}$ . Once the binning is done, we apply the following statistical analysis methods. They are similar to the three presented above, but applied to the 10 bin values rather than 2D data fields.

- Average Difference of Averages of the bins (ADA of bins): we compute the average value of the bins of each ensemble member and of bins of the target, and compute the absolute difference between the two. This method do not leads

to exactly the same results as the ADA without binning because some bins might contain a different number of grid cells than others, meaning that they cover slightly different areas but still get exactly the same weighting in the binning approach. As we intend here to compare methodologies, we consider relevant to also test this method, even if it gives results very close to ADA without binning, to quantify how much they differ.



- Root Mean Square Error of the bins (RMSE of bins): we compute the RMSE value between the bins of each ensemble member and the bins of the target.
- Mean Absolute Error of the bins (MAE of bins): we compute the MAE value between the bins of each ensemble member and the bins of the target.

Figure 1. Schematic showing how the binning of melt rates is done.  $\bar{B}_x$  stands for the value of each bin which is the average value of 10% of the ordered values. The darker and lighter shadings represent each 10% of the corresponding dataset. The panel (a) is the distribution of the values of the target, the panel (b) is the distribution of the values of the first ensemble member, and the panel (c) is the one of the last ensemble member. Once the bins for the target and all the ensemble members are calculated we apply the statistical methods to rank the ensemble members.

For all of the methods above, the ranking of the best ensemble members is given by the lowest values obtained with the given equations in the Table 1. The results of the analysis applied to all ice shelves of the Antarctic ice sheet are given in subsection

3.1. We also apply the above methods at a more local scale than Antarctica to test whether a local calibration is needed. To accomplish this we apply the above methods to two additional areas of the Antarctic ice sheet: the Filchner-Ronne ice shelf (FRIS), and the sector of the Bellingshausen and Amundsen seas (BA seas). The results are given in subsection 3.2. To test the robustness and sensitivity of the methods under more conditions we also test them with different resolutions of the ice sheet model, ocean temperature and salinity forcings, and targets of basal melt rates. We discuss these results in section 4.

#### 2.6 Future applications: ISMIP 2300

To make a preliminary assessment of the relevance of this implementation and of the calibration methods, we run a small ensemble of future scenarios. For these simulations, the PICO parameters values are consistent with the results of the analysis of the calibration ensemble defined in sub-section 2.5 and presented below in Figure 2. We make use of the ISMIP6 2300 protocol in which GRISLI and other ice sheet models using PICO participated (Seroussi et al., 2024). The basal melt parameterisation used in the submitted GRISLI simulations for the ISMIP 2300 was the quadratic non-local melting parameterisation from Jourdain et al. (2020) which, in the following, will be referred to as QuadNL. For these simulations with PICO we do not recalibrate GRISLI mechanic parameters and use the same initial state as for QuadNL. Here we repeat the experiment "expAE05" from ISMIP 2300 which corresponds to climate forcing computed by the UK Earth System Model (UKESM1-0-LL) for the scenario SSP5-8.5 (Seroussi et al., 2024). With our new simulations we will assess: i- the difference of sensitivity between PICO and QuadNL with the same model and same forcings; ii- the importance of calibration choices on model results and; iii- how the response of GRISLI-PICO differs from other ISMIP participating models that also used PICO. The results are presented in subsection 3.7.

#### 3 Results






#### 3.1 Can we capture the spatial or binned distribution of melt rates using any of the six calibration methods?

Here we present the results for the six calibration methods explained above in section 2 applied to all ice shelves of Antarctica at a 16 km resolution. The main calibration results for each method are presented in the Figure 2, more detailed results are shown in the supplementary materials section 1. In the Figure 2, we show in panels (a), (b), (c), (g), (h), and (i) the distribution of the values of the five best ensemble members according to each methods that we can compare directly with the distribution of the observations from Adusumilli et al. (2020). In panels (d), (e), (f), (j), (k), and (l) we show on the heatmaps the ranking of all the ensemble members for each corresponding method. We also highlight the five best members with the black dots numbered of their member number.

Panels (a) and (d) show the results using the ADA method. We see that the best members using the ADA method cover a large range of values from  $0.1 \times 10^{-5}$  m s<sup>-1</sup> to  $3.0 \times 10^{-5}$  m s<sup>-1</sup> for  $\gamma_T^*$  and from 0.1 Sv m<sup>3</sup>.kg<sup>-1</sup> to 5.0 Sv m<sup>3</sup> kg<sup>-1</sup> for C. Also, the matching with the target distribution of the top five members is in some cases good (members 36 and 35) and in other not (members 133, 56 and 44). The large spread of the top five members demonstrate that with this method the best parameters

will not systematically be of the same order of magnitude of values and can depend heavily on the sampling ensemble. The results for the 2D RMSE and 2D MAE methods are shown in panels (b), (e) and (c), (f), respectively. Both methods gives quite similar responses, the top five members for both methods are side-by-side. In comparison to the ADA methods, these two methods enable us to have a narrower range of PICO parameter values, in particular for the parameter  $\gamma_T^*$ . However, none of the selected top five members, for both methods, matches the distribution of the observations (panels (b) and (c)).




The following panels, from (g) to (l), correspond to the results of the methods using binning. The first of the three, the ADA of bins (panels (g) and (j)), selects almost the same members as the ADA without binning. Therefore, using binning before computing the ADA is not enough to have a small set of best members and fitting the distribution.

The next methods, RMSE of bins and MAE of bins, minimise the differences between the bins of the ensemble members and the bins of the target systematically. Because of this we obtain a selection of best ensemble members that systematically fit the distribution of the target (panels (h) and (i)). But also the range of parameters corresponding to the best five members is small, all the top members are side-by-side (panels (k) and (l)). This gives us confidence that these two last methods consistently give the same range of parameter values for different ensemble sampling that would also matches best the distribution of the target.

Figure 2. Comparison of the 6 methods. The panels (a), (b), (c), (g), (h) and (i) are the results of the distribution of the best five members according to each methods. The panels (d), (e), (f), (j), (k), and (l) show the ranking of all the ensemble members according to each methods. The red square shows the best member. The black dots with numbers show the best five members for each method, the same members as on the distribution panels.

The inability of ADA of bins to give a small range of parameter values can be explained by the fact that it allows compensation between bins. Figure 3 panels (a), (c), (e), show the values of the 10 bins of the five best members according to the three methods using bins. We see a compensating effect between the lower and higher parts of the values of the bins, in particular for the members 133, 56 and 120. In other words, these members score well with this method because, in this case, the positive differences to the target in the lower bins is compensated by the negative difference to the target in the higher bins. Whereas, the RMSE of bins and the MAE of bins have systematically smaller anomaly values and do not allow for compensating effect.

The largest discrepancies between the binned values of modelled melt rates and those from observational datasets occur at the extremes, that are the bins with the lowest 10% and highest 10% of the values from the distribution. We therefore analyse further these two bins for all the PICO configurations in panels (b) and (d) of Figure 3, respectively. These panels reveal distinct sensitivities to the two PICO parameters. For example, at fixed values of C, increasing  $\gamma_{\rm T}^*$  consistently raises the bin values in both the lowest (panel (b)) and highest (panel (d)) deciles. This implies a reduction in error when the model underestimates melt (negative error, in blue), or an amplification of error when it overestimates melt (positive error, in red). In contrast, at fixed  $\gamma_{\rm T}^*$ , varying C can produce divergent effects between the lowest and highest bins. For instance, at  $\gamma_{\rm T}^* = 2.0 \times 10^{-5}$  m s<sup>-1</sup>, increasing C leads to a decrease of bin error values in the lower 10% bin (b) and an increase in bin error values in the upper 10% bin (d). Finally, by computing the sum of the absolute values of the errors for all 10 bins we can find the combinations of PICO parameters that minimize this error the most. The panel (f) of Figure 3 shows the results with superposed the best members according to the MAE of bins methods shown on panel (e). We find that the two metrics, MAE of bins and sum of absolute errors of the bins, leads to a similar selection of the best ensemble members.

Figure 3. Anomaly of the values of the bins for the five best members according to each methods, (a) ADA of bins, (c) RMSE of bins and (e) MAE of bins, with regard to the target Adusumilli et al. (2020). The closer the value is to 0, the closer value of the bin of the ensemble member is to the value of the bin of the target. Panels (b), (d) and (f) show the error in percentages between the model result and the observations for the lowest bin (b), highest bin (d), and the sum of of the absolute error of the 10 bins (f). The values of the bins are the same for all three methods using binning, therefore results shown in panels (b), (d) and (f) are relevant for the three methodologies. The black dots with numbers on panel (f) show the five best members according to the MAE of bins, and the red square shows the best member for this same method.

Nonetheless, being able to match the distribution can also mean spatial compensation between different locations. Therefore we look at the spatial distribution of the values on the Figure 4. It shows the single best member of each methods corresponding to the red square on the heatmaps of Figure 2. We see that four methods (ADA, ADA of bins, 2D RMSE, and 2D MAE) results in a spatial distribution with little contrast between higher and lower values, they do not even have values more negative than  $-1 \text{ m.yr}^{-1}$  in blue (Figure 4 panels (a) to (c)). It could be because this selection gives low  $\gamma_{\rm T}^*$  values,  $0.1 \times 10^{-5} \text{ m s}^{-1}$  and  $0.25 \times 10^{-5} \text{ m s}^{-1}$ . Whereas, the best single member following the RMSE of bins or the MAE of bins have a lot more contrast (Figure 4 panels (d) and (e)), which corresponds better to what is seen in the observations (Figure 4 panel (f)). These two methods results in higher  $\gamma_{\rm T}^*$  values:  $1.5 \times 10^{-5} \text{ m s}^{-1}$  and  $2.0 \times 10^{-5} \text{ m s}^{-1}$ .




Figure 4. Spatial distribution of the basal melt rate values for the six calibration methods tested ((a) to (e)), N.B. (a) represent two methods (ADA and ADA of bins) as they give the same best ensemble member. (f) is the spatial distribution of the observations from Adusumilli et al. (2020). In panel (f), the rectangles show the two chosen areas to test the calibration at two smaller scale than Antarctic wide and presented in sub-section 3.2, the red further to the left is the Bellingshausen and Amundsen seas (BA seas) area and the blue further to the right the Filchner-Ronne ice shelf (FRIS) area.

Overall, we see that calculating the average, with or without binning, can lead to very different optimal PICO parameter values, which can be explained by the possibility that there is compensation between negative and positive values. The four other methods shows more systematic results where the best members points are all side-by-side. However, between the methods

without (2D RMSE and 2D MAE) and with binning (RMSE of bins and MAE of bins) the selected members are in different parameter spaces. By observing the spread of the rankings combined with the distributions, we can consider the methods RMSE of bins and MAE of bins as the best ones among the six tested here. This is justified by the fact that the selected members are: i) better able to match the distribution curve from the target (Figure 2, panels (a), (b), (c), (g), (h), (i)), ii) systematically give best values in the same small range of values (Figure 2, panels (d), (e), (f), (j), (k), (l)), and iii) the magnitude of the spatial patterns is similar to the target (Figure 4).

#### 3.2 Do we need to calibrate the PICO parameters locally?




Here we show how different the results would be if we would calibrate PICO for a specific domain of the Antarctic ice sheet, and we assess whether a Antarctic wide calibration is suited to domain-wide applications. Figure 5 presents a selection of the analysis similar to the previous section, but applied to the two domains: BA seas on the left side, and FRIS on the right side. The results shown here correspond only to the method MAE of bins considered as one of the two best methods, results for the other methods can be seen in the supplementary materials sections 2 and 3. First, with panels (a) and (b) we see that it is more challenging to match the observation distribution in the BA seas than for the FRIS. But also, on these two panels we also show the best Antarctic-wide calibration following the MAE of bins. In the BA seas, the top member, 34, is the same as for the Antarctic wide selection. For FRIS, the best Antarctic wide selection is also part of the top five members. Second, with panels (c) and (d), we see a strong difference between the two sectors in the sensitivity of the average basal melt to change of the PICO parameter values. A change of the overturning coefficient of  $+0.4.10^{-5}$  m s<sup>-1</sup> would lead to an average basal melt value higher than 4 m yr<sup>-1</sup> above the average of the target in the case of the BA seas sector, whereas it would barely make any difference for the FRIS sector. Thus, the best Antarctic wide calibration using the MAE of bins is similar to the best possible local calibration, with negligible differences in less sensitive areas.

Figure 5. Comparison of the calibration between the Bellinghausen/Amundsen seas sector (on the left side) and Flichner-Ronne ice shelf sector (on the right side). (a) and (b) are the distribution of the five best members for both sectors using the MAE of bins methods, with the distribution of the best calibration applied Antarctic wide (in purple). (c) and (d) show the absolute difference between the average of each of the 169 ensemble members and the average of the observations. However, the five best members shown with the numbered black dots correspond to the best members with the MAE of bins method (as in (a) and (b)). The best member for the local and Antarctic wide calibration using the MAE of bins is given by the red square and the purple hexagon respectively.

#### 3.3 How robust are the methods if applied at different resolutions and areas of the ice sheet?


To test how robust the results are, we run an additional calibration ensemble of 169 members with a resolution of 40 km. We use the same forcing from Jourdain et al. (2020) and target from Adusumilli et al. (2020) regridded at 40 km. With this new calibration ensemble we renew the analysis for Antarctica wide as well as for BA seas sector and FRIS. We therefore test each statistical methods with six different cases: two resolutions  $\times$  three areas of interests. The detailed analysis are shown in supplementary materials for all six conditions in supplementary materials sections 3 to 8. We summarise all these conditions

by aggregating the top five members in all the six conditions for each statistical methods and plot them in Figure 6. It enables us to visualise the spread of the top PICO parameters aggregated over the different cases. We can see that the methods RMSE of bins and MAE of bins are the two methods that gives consistent optimal PICO parameter under all different conditions. But also they give more systematically the same members (see appendix A1), suggesting to use the same order of magnitude of the parameters for all the tested conditions. This consistency can matter to inter-compare: i) different parameterisations; ii) when the same parameterisation is used in different ice sheet models; iii) results at different resolutions. No clear trend can be seen between the best members at 16 km and 40 km resolution (see appendix A2). It is particularly relevant to have robust methods over different resolutions to make the comparisons more systematic such as in the scope of Ice Sheet Model Inter-comparison Projects (ISMIP) (Seroussi et al., 2024), Marine Ice Sheet-Ocean Model Intercomparison Project (MISOMIP) (Rydt et al., 2024), or comparison between paleo and future ice sheet behaviours (Golledge et al., 2021).


Figure 6. Average ranking of each calibration method tested under two different resolutions (40 km and 16 km) and applied to three different sectors (Antarctic wide, BA seas, and FRIS). The top five members of all six conditions are shown with the black and white hexagons.

#### 3.4 How sensitive are the methods to the forcings?




In all presented results until here we used the forcing from Jourdain et al. (2020). Here, we assess how sensitive the results are to this forcing. We run three additional calibration ensembles of 169 members at 16 km resolution, using the same forcing by Jourdain et al. (2020), but with a different temperature correction on the top of it. The temperature corrections are (1) + 1K, (2) + dT from Reese et al. (2023) (see their Table S1), and (3) + dT from Jourdain et al. (2020) (see their Figure 5 panel (a) for the quadratic non local parameterisation). Reese et al. (2023) and Jourdain et al. (2020) apply a temperature correction that differs per drainage regions. To remain concise, in Figure 7 we only present the results of the ranking for the MAE of bins method (for additional analysis see supplementary materials section 9, 10 and 11). We can observe that with a correction of +1 K both PICO parameters shift to slightly lower values. The opposite is happening with the correction from Reese et al. (2023), which is expected since the temperature are almost all negative values, reaching up to -2 K. The correction from Jourdain et al. (2020) makes only minor differences as the values are rather low with a maximum absolute value of +1.07 K. These results suggest that using warmer forcing for the calibration will lead to a calibrated PICO less sensitive to temperature changes, and vice versa with a colder forcing. Finally, we can state that, even after combining all the ranges of values suggested by the four different forcings, forcing uncertainties lead to a smaller range of PICO parameter values than using different statistical calibration methods. In other words, the choice of the calibration method is more important than the choice of the forcing. Since the best parameters do not vary much (about - 0.5 for the  $\gamma_T^*$  and - 0.05 for C) between the + 0 K and the + 1 K forcings, we analyse in the next subsection what is the sensitivity of PICO to understand better what would be the response of PICO to warmer than present-day conditions.

Figure 7. Ranking using the MAE of bins to all Antarctic ice shelves with different forcings. (a) is without additional temperature correction, (b) is with +1 K, (c) is with dT defined by Reese et al. (2023), and (d) is with dT defined by Jourdain et al. (2020).

#### 3.5 What is the melt rate sensitivity of PICO to changes in ocean temperature?




Thanks to the ensembles with + 0 K and + 1 K ocean forcings from the previous subsection we can determine the melt rate sensitivity of PICO to changes of ocean temperature for all the ensemble members. We are here assuming a linear sensitivity and therefore compute it as the difference between the + 1 K experiment minus the + 0 K experiment. The results are shown in Figure 8 where we also differentiate the sensitivity in the 3 areas defined for the analysis shown in subsection 3.2. First of all, we see that in most cases the sensitivity of PICO increases when the value of either of the two parameters is increased. Second, we see that the sensitivity varies between areas. For instance, 1 K of warming with the same PICO parameters as used by PISM-PICO (Seroussi et al., 2024) (squares in Figure 8) would lead to an increase of 1.5 m yr<sup>-1</sup> K<sup>-1</sup> in FRIS, whereas the increase would be 8.4 m yr<sup>-1</sup> K<sup>-1</sup> in the BA seas ice shelves. We also see that the range of possible sensitivities (panel (d)) is about four times larger in the BA seas than in the FRIS. These results quantify how much the sensitivity of the basal melt rate would change, globally and regionally, by changing the values of the two PICO parameters. In all the cases, the best calibration with the MAE of bins (black hexagons) is in the lower range of all the tested combinations of parameter values. These values are also lower than the range of Antarctic ice shelves sensitivity estimates from some previous studies (Levermann et al., 2020;

van der Linden et al., 2023), but closer to the PICO sensitivity obtained by Reese et al. (2023); Lambert and Burgard (2025) when optimising parameters for present-day melting. The methodologies in the assessments of the sensitivities are however different in each study. Nonetheless, based on the results show in Figure 8 we can expect a low to moderate response of ice shelves in this calibrated version of PICO to future projections scenarios.

Figure 8. Linear sensitivity of PICO for the 3 areas of interest: (a) all ice shelves of Antarctica, (b) Flichner-Ronne ice shelf and (c) Bellingshausen and Amundsen seas ice shelves. Panel (d) compares the ranges of sensitivity of the three areas. In all the panels, the hexagon represents the calibration done in this study with the MAE of bins, the square represents the calibration used in PISM-PICO for ISMIP 2300 (Seroussi et al., 2024).

#### 3.6 How sensitive are the methods to the target?

As presented in subsection 2.4, we took as target the basal melt rates retrieved from Adusumilli et al. (2020). However, the retrieval of basal melt rates from satellite observations is poorly constrained. Hence, we could make an argument for choosing a different target for the calibration of the basal melt rate parameterisation. Therefore, to assess the uncertainty due to the choice of the melt rate target, we ran the same robustness analysis as in section 3.3, but with as target the basal melt rates retrieved by Paolo et al. (2023) instead of the target of Adusumilli et al. (2020). The detailed results are shown in appendix B. Overall, they are similar to the ones obtained with Adusumilli et al. (2020) as target, with a slight shift towards higher  $\gamma_T^*$  values. We also make the bins analysis done in Figure 3 with the datasets from Paolo et al. (2023) and we observe similar results (see Supplement Figure S29). This is expected as the distribution and the main statistics are very similar (see Figure 9 panel (a), with a higher standard deviation for the Paolo et al. (2023) dataset). Figure 9 panel (b) shows the spatial differences between the two datasets. The magnitude of these differences is of the same order of magnitude as the values of basal melt rates themselves, reaching values below -4 m yr<sup>-1</sup> and above 4 m yr<sup>-1</sup>. This observation is made for both resolutions 16 km and 40 km (see supplementary materials section 13). The standard deviation of the difference between the two datasets is 1.97 m yr<sup>-1</sup>, which is about three time larger the mean melt rate of both datasets. Hence, despite the important spatial differences, we conclude that the calibration methods are not significantly sensitive to the choice of the target dataset as they both give a similar selection of best ensemble members for each method.

Figure 9. (a) Histogram of values of the two datasets and main statistics. (b) Spatial distribution of the difference of basal melt rates between Adusumilli et al. (2020) and Paolo et al. (2023). Results on this Figure are shown for the mesh grid resolution of 16 km, more details are given for mesh grid resolution of 16 km and 40 km in supplementary materials section 13. The difference of the means  $(0.66 - 0.68 = -0.02 \text{ m yr}^{-1})$  is different from the mean of the difference  $(0.16 \text{ m yr}^{-1})$  because in the mean of the difference only the grid cells with values in both datasets are taken into account.

#### 3.7 How much does the calibration method matter for future projections?



Following the previous analysis, we select seven cases to make a first-order assessment of the impact of the calibration method choice on future projections of the Antarctic ice sheet until 2300 (Figure 10 (a)). Five of the seven cases correspond to the best members according to the six methods applied Antarctic-wide. We get five cases rather than six because ADA and ADA of bins give the same best member. The two additional members correspond to the parameter values chosen by the ice sheet models PISM/Elmer-ice (101) and Kori (102) that use PICO for the ISMIP 2300 experiment Seroussi et al. (2024). Figure 10 panel (b) shows the distribution of the basal melt rate at the very start of the simulations for the ISMIP 2300 simulation for the year 2015, compared to the observations from Adusumilli et al. (2020). For comparison, we include GRISLI with the QuadNL parameterisation (Jourdain et al., 2020).

Figure 10. (a) Selection of PICO parameter values for ISMIP2300 applications in this study based on the different calibration methods and previous PICO calibrations. (b) comparison of the basal melt rate between Adusumilli et al. (2020) and the start of the simulation (t=2015) for each calibration.

The main results of this small ensemble of ISMIP 2300 members are shown in Figure 11. In Figure 11 panel (a) we see the total basal mass balance flux (BMB flux) over time. We can see very different behaviours between ice sheet models and parameterisations. Over the whole simulation the highest values are obtained with the QuadNL parameterisation with GRISLI and Kori. Then in the medium range we have the calibrations 101 and 102 of PICO with different ice sheet models (Elmer-ice, PISM, Kori and GRISLI). In the upper part of the figure, the group of simulations with the lowest basal melt rates corresponds to all the GRISLI-PICO simulations with the different calibration methods done above. In Figure 11 panel (b), the floating ice area representing the size of the ice shelves show significant differences between the different simulations, including the GRISLI-PICO with different calibrations. The growth of the floating ice of the GRISLI-PICO calibrations 101 and 102 is something observed also in the Kori-PICO simulation. The calibration 34, considered as the best one in the calibration process, shows a small decreasing trend in floating ice area over the course of the simulation compared with most other cases. Finally, Figure 11 panel (c) shows the contribution to sea level rise of all the simulations. We can see that simulations with PICO with


the same PICO parameters lead to lower values of sea level contribution by 2300 for both GRISLI and Kori compared to results from the same ice sheet models but using the QuadNL parameterisation. Elmer-ice with PICO even suggests a negative sea level contribution from Antarctica throughout the simulation. Except compared to Elmer-ice with PICO, all the simulations of GRISLI-PICO result in lower values of sea level contribution by 2300 than all the other cases.

Figure 11. ISMIP 2300 applications with different PICO calibrations. (a) shows the evolution of the total basal melt balance (BMB) beneath floating ice over time. (b) shows the evolution of the floating ice area. (c) shows the contribution to sea level rise.

#### 4 Discussion







Here, we develop the paper further by discussing three questions: What can be the right observational target? What can we do to better understand the sensitivity of the Antarctic ice shelves to warming oceans? And why does PICO lead to lower sea level contribution estimates than QuadNL in GRISLI?

#### 4.1 What could be the right observational target?

The observation of the disagreement between the two target datasets shown in Figure 9 is important to justifying the usage of bins suggested in the present study. Indeed, by using the binning methods we give spatial freedom to the datasets and constrain them by their values. Since calibration methods that minimise cell-to-cell differences between modelled and observed melt rates often fail to capture the overall distribution of observed values, and given the spatial inconsistencies among observational datasets, we prioritise reproducing the correct distribution of basal melt rates over minimising spatial mismatches. We consider that having a good statistical representation of the melt rates is potentially more important for the dynamic of the ice-sheet. For instance, the highest melt rate values are observed in the Amundsen sea area, where due to the retrograde slope the West Antarctic ice sheet is exposed to the marine ice sheet instability process (Weertman, 1974; Joughin et al., 2014), therefore capturing these high melt rates values is potentially important for future projections. Moreover, even if we might not have the right values at the right locations within a given ice-shelf, we have seen in subsection 3.2 that the calibration method MAE of bins enable to have the values close to the distribution of the target in local areas. Prioritizing values over spatial correspondence within an ice shelf is in agreement with Joughin et al. (2021) who argue that the ocean-induced melt volume, regardless of the spatial distribution, directly paces the ice loss. However, other studies suggest that localized sub-ice-shelf melt can have a strong impact on the buttressing or that in more strongly buttressed areas sub-ice-shelf melt would have outsized effect (Gudmundsson, 2013; Reese et al., 2018b). Additionally, one limitation here is that by scaling up the resolution of the observational datasets to the ice sheet resolution, 16 km or 40 km here, we are losing most of the data points with melt rates values above 6 m yr<sup>-1</sup> (see supplementary materials section 13). Consequently, certain ice shelves in the West Antarctic ice sheet such as Thwaites or Pine Island have very few grid cells at 40 km resolution. This can be an important limitation, yet we consider it important to calibrate and test for this coarse resolution because it is an option for paleo ice sheet simulations. This also means that a calibration by using the same method but for higher resolution models or with irregular grids might have different values of the PICO parameters than the ones found here. At a higher resolution we could also consider computing more than 10 bins, this has not been explored in this study.

Nonetheless, with this analysis using the observational dataset from Paolo et al. (2023) we can justify that the MAE of bins method is more robust than the RMSE of bins in a case where we want to use one set of parameters for the all Antarctic ice shelves. Indeed, as shown in Figure 12, the range of values of the parameters suggested by the top five members is smaller for the MAE of bins (C ranging from 0.01 to 0.25 Sv m<sup>3</sup> kg<sup>-1</sup> and  $\gamma_{\rm T}^*$  from  $1.00 \times 10^{-5}$  to  $5.00 \times 10^{-5}$  m s<sup>-1</sup>) than the RMSE of bins (C ranging from 0.01 to 0.50 Sv m<sup>3</sup> kg<sup>-1</sup> and  $\gamma_{\rm T}^*$  from  $1.00 \times 10^{-5}$  to  $15.00 \times 10^{-5}$  m s<sup>-1</sup>).

Figure 12. Average ranking of two calibration methods, RMSE of bins (a) and MAE of bins (b), tested under two different resolutions (40 km and 16 km) and applied to three different sectors (Antarctic wide, BA seas, and FRIS) with target the observational dataset from Paolo et al. (2023). The top five members of all six conditions are shown with the black and white hexagons.

#### 4.2 Towards a better understanding of the sensitivity of Antarctic ice shelves to warming oceans

Understanding the sensitivity of the Antarctic ice shelves to ocean warming is key to be able to make future projections. In subsection 3.5 we quantified the Antarctic ice shelves sensitivity of our PICO version, based on a highly simplified approached. More advanced methods have been developed, for instance, Lambert and Burgard (2025) apply both salinity and temperature anomalies that compensate each other to maintain a present-day like density profile. Previous studies report a wide range of sensitivity estimates, spanning up to an order of magnitude (Levermann et al., 2020; van der Linden et al., 2023; Lambert and Burgard, 2025), however they differ in their methodology to compute the sensitivity. These differences include the choice of forcing, the sub-shelf melt parameterisation, model resolution, and whether a linear or quadratic sensitivity is assumed. While this goes beyond the present study, developing a standardized framework for quantifying the sensitivity of the ice shelves

to ocean warming would be useful to facilitate the comparisons across studies and models. In parallel, continuing efforts in Earth observations to extend and refine records of ocean properties and sub-shelf melt rate remain critical to improving the robustness of sensitivity estimates. Additionally, we observe a large range of possible melt rate sensitivities to ocean properties with changes in the values of the parameters (Figure 8). Therefore, we could argue for calibrating basal shelf melt rate parameterisations directly to a target value of sensitivity. This has been done by Reese et al. (2023), leading to higher PICO sensitivity than the previous calibration (Reese et al., 2018a), and used in the future projections until 2300 (Seroussi et al., 2024).

#### 425 4.3 Why does PICO lead to lower sea level contribution estimates than QuadNL in GRISLI?





Overall, including PICO in ice sheet models leads to lower sea level contribution in the simulations up to 2300 shown in Figure 11. Indeed, Kori with the PICO parameterisations produces 1.5 m sea level equivalent less than with the QuadNL parameterisation. Similarly, all the PICO calibrations with GRISLI have a lower sea level contribution than with the QuadNL parameterisation. Elmer-ice with PICO suggests even a negative sea level contribution from Antarctica all along the simulation. Of course, there are many other factors influencing future sea level predictions in ice sheet models that are not related to the parameterisation of the ice-ocean interactions; and it is definitely possible to have larger sea level contribution with PICO as shown in the simulation with PISM. But here we want to provide some hypotheses that could explain this overall pattern of lower sea level contribution from simulations with PICO. We outline five possibilities:

- The overturning circulation under the ice shelves, which tends to reduce the basal melt rate, is computed differently in the parameterisations PICO and QuadNL. In PICO the overturning fluxes is computed with the overturning circulation coefficient C and the difference of densities (see equation 2). Whereas in the QuadNL it is in the product involving the thermal forcings which results in stronger overturning from warmer conditions (see equation 1 in Jourdain et al. (2020)).
- PICO includes a linear relationship between the ocean temperature and the basal melt rate for high temperatures, whereas
   QuadNL has a quadratic one. It means that for high projections, if both start off with a similar basal melt rate, the QuadNL will project significantly higher melt rate values with increasing temperatures.
- The QuadNL parameterisation takes as input 3D fields of oceanic forcings. Whereas PICO takes one value per ice-shelf  $(T_0 \text{ and } S_0)$  which is an average of values over the continental shelf at the depth of the continental shelf, in front of the concerned ice-shelf.
- PICO tends to have more smoothed out melt rates and does not show significantly higher melt rates at grounding lines, as seen in satellite-derived fields. If the basal melt rates at the grounding line is a major factor for ice loss, it could explain a less sensitive response.
  - The sensitivity to ocean warming in our calibrated version of PICO lies below some previously reported ranges of Antarctic ice shelf sensitivity (Levermann et al., 2020; van der Linden et al., 2023), and is more consistent with the PICO sensitivity range estimated by Lambert and Burgard (2025). In their study, the PICO sensitivity is lower than

that obtained with other sub-shelf melt rate parameterisations, although it is not the lowest overall. By contrast, the QuadNL parametrisation lies on the higher end of sensitivity spectrum compared to other parametrisations (Burgard et al., 2023; Lambert and Burgard, 2025). So far, neither PICO nor QuadNL sensitivities ranges, which also depend on their calibration, can be ruled-out as we do not know what the right sensitivity is.

In addition, the results of the future simulations show that the ice shelves can have different behaviours depending on the calibration method and the choices of the values of the parameters. Thus, we here advocate for a calibration methodology that best fits the full distribution of the observational datasets, it is a more physical calibration of the process modelled than simply matching the average value for instance. This methodology can potentially be applied to modules in other models that benefit from existing observational datasets. However, regardless to the quality of the calibration, all parameterisations are simplifications of processes. Therefore, one should be aware of the processes represented or not in a model for the interpretation of the outputs, and when possible still run an ensemble to cover the range of uncertainties.

#### 5 Conclusions and perspectives





The Antarctic ice sheet retreat is driven by ice-ocean interaction and differences in the ice-ocean parameterisations can lead to major differences in simulations of future dynamics of the ice sheet. We presented the implementation of the PICO basal ice shelf melt module (Reese et al., 2018a) in the GRISLIv.2 ice sheet model (Quiquet et al., 2018). Then we compared six statistical calibration methods to find the best set of two PICO parameters C and  $\gamma_T^*$ . We demonstrated that the only two methods, the RMSE of bins and the MAE of bins that also fit the low and high extremes in the target histogram, provide a robust constraint of both parameters in a narrower range of values. They give more consistent results, making them more reliable and less dependent upon the ensemble sampling. The results from these two methods also better reproduce the range of special variability, if not the details of spatial patterns observed in the chosen target (Adusumilli et al., 2020). By using these two methods that closely fit the entire distribution of the target for all Antarctic ice shelves combined, we also show that region-specific calibration is not necessary. According to this research, the best values of the PICO parameters in our specific set up are  $\gamma_T^* = 2.0 \times 10^{-5}$  m s<sup>-1</sup> and C = 0.1 Sv m<sup>3</sup> kg<sup>-1</sup>.

We did future simulations without re-calibration of the mechanical parameters of GRISLI, such as ice flow and drag, to have a preliminary assessment of the impact of the choice of the calibration method applied to PICO. On the one hand, we see that the re-calibration of the PICO parameters can lead to major differences in sea level contribution compared to simulations using parameters values used in previous studies. We showed that only our calibration fits the whole distribution of sub-shelves melt rates from the observations. On the other hand, the choice of the calibration method does not have a major direct impact on the sea level contribution, yet it does have a significant impact the ice-shelf extent. This remains coherent as the relationship between ice-shelf changes and sea level contribution is still uncertain and ice sheet model dependant (Sun et al., 2020). Thus, our analysis of future simulations is not enough to gain confidence in one specific projection, further work is needed to constrain the sensitivity of the Antarctic ice sheet using paleo records and coupled ice sheet-ocean models.

Finally, we thoroughly tested the statistical methods by assessing how robust the results are by applying them to additional cases such as different resolutions, regions of Antarctica, forcings, and targets. This assessment give us confidence in our results confirming that the RMSE of bins or the MAE of bins methods are the most robust ones and could avoid the need for modellers to use temperature corrections on top of the parameterisation as well as give more confidence in paleo ice sheet applications at lower resolutions using present-day data for the calibration (Quiquet and Roche, 2024). The principle of using bins is justified by observing the magnitude of the spatial disagreement between the observational datasets (Adusumilli et al., 2020; Paolo et al., 2023). As the MAE of bins gives a smaller parameter space under the six conditions and different targets tested, we recommend using this method.


To progress further, we invite ice-ocean interaction modellers to test the MAE of bins method in their own set up of ice-ocean parameterisation, ice sheet model and initial state. Another possible improvement would be to target a sensitivity of the basal melt rate to ocean forcing changes rather than targeting a given basal melt rate for a given ocean forcing. Reese et al. (2023) calibrated PICO to a sensitivity to temperature, but it required the use of temperature corrections. We suggest that using the MAE of bins calibration method could enable calibrating to sensitivity without additional temperature corrections. Finally, when possible, we encourage modellers to quantify the sensitivity of their sub shelf melt parameterisation.

Code and data availability. The GRISLI model with the PICO implementation, the outputs of the simulations, as well as the Jupyter Notebook files to do the figures are available on Zenodo (Menthon et al., 2025).

Appendix A: Standard deviation of the rankings for each methods applied to 6 different conditions (two resolutions × three areas of the Antarctic ice sheet)

Figure A1. Same as Figure 6 but showing the standard deviation of the rankings under the six different conditions instead of the average. It gives additional information about the degree of confidence and robustness of the methods.

Figure A2. Same as Figure 6 but differentiating the 2 resolutions.

# Appendix B: Average and standard deviation of the rankings for each methods applied to six different conditions (two resolutions $\times$ three areas of the Antarctic ice sheet) with as target Paolo et al. (2023) instead of Adusumilli et al. (2020)

Figure B1. Same as Figure 6 but with target Paolo et al. (2023)

Figure B2. Same as Figure 6 but the standard deviation with target Paolo et al. (2023)

Author contributions. MM led the project and performed the majority of the work. MM made the PICO development and implementation in GRISLI with contributions from PB, AQ and DMR. MM, PB, AQ and RR designed the simulations. MM ran the simulations and analysed the results. MM wrote the manuscript with comments and contributions from all co-authors. PB, AQ and DMR supervised the project. PB acquired the fundings.

Competing interests. The authors declare that they have no conflict of interest.

Disclaimer: TEXT


Acknowledgements. The authors thank the two reviewers, Xylar Asay-Davis and Clara Burgard, for their comments that enabled to significantly improved the paper. Maxence Menthon was supported by NWO Grant OCENW.KLEIN.243. All the colour-bars in the figures are using Crameri et al. (2020) colour schemes, and the time series are marked with distinguishable markers, to ensure the best accessi-

bility. For the future projections, we acknowledge CMIP6 and ISMIP6 23rd Century Projections participating modeling groups, and the ESGF centers (see details on the CMIP Panel website at https://wcrp-cmip.org/cmip-overview/). Original forcings data sets and simulations results for two-dimensional fields are available on Ghub (https://theghub.org/dataset-listing) under "ISMIP6 23rd Century Forcing Data sets": https://theghub.org/resources/5161 and https://zenodo.org/records/13135571 (Nowicki and ISMIP6 Team, 2024a) (Nowicki & ISMIP6 Team, 2024a). Model outputs from the ISMIP-2300 simulations are available on Ghub (https://theghub.org/dataset-listing) under "ISMIP6 23rd Century Projections": https://theghub.org/re- sources/5163 and https://zenodo.org/records/13135599 (Nowicki and ISMIP6 Team, 2024b).

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
