# Peer review of "Comparison of calibration methods of a PICO basal ice shelf melt module implemented in the GRISLI v2.0 ice sheet model"

_EGUsphere, 2025_

## Referee Comment (RC1)

**Review of Menthon et al. "Comparison of calibration methods of a PICO basal ice shelf melt module implemented in the GRISLI v2.0 ice sheet model"**

Reviewer: Xylar Asay-Davis

I wish my name to be relayed to the authors, as I feel I am always a better reviewer when I am not anonymous and I encourage others to consider reviewing non-anonymously whenever they feel able.

**General Comments:**

This paper presents the implementation of the PICO parameterization for Antarctic basal melting in the GRISLI v2.0 ice sheet model. Then, the authors analyze a series of potential methods for parameter calibration, settling on a binning technique together with the mean absolute error (MAE) as the most robust approach. They probe the robustness of the approach in several ways, including modifying the ocean forcing, changing melt-rate dataset use as the observational target, calibrating regionally rather than Antarctica-wide, and using different ice-sheet resolutions (also the resolution of the parameterization). A significant finding is that their calibration method requires no correction of the input ocean temperature field, in contrast to previous work. In addition to the calibration, they ran an ensemble of projection simulations to the year 2300, showing the impacts on total melt flux, floating ice area, and Antarctic mass change from different PICO calibrations.

The paper is quite detailed and for a fairly niche audience. Even so, I think the level of detail is justified because the analysis performed here is potentially applicable not just to the PICO parameterization but to all Antarctic basal melt parameterizations. The results present a strong case that the binning technique used for calibration leads to parameter choices that are robust to the various uncertain inputs to the parameterization and which represent the statistics if not the detailed spatial patterns of Antarctic basal melting.

I see a potentially significant issue in the equation presented for the pressure at the ice shelf-ocean interface that could impact the results. As I expand on in my specific comments below, the pressure as presented is computed from the ice density but the ice draft (the ocean thickness, so to speak), whereas the correct pressure must involve either the ocean density and the ice draft or the ice density and full ice-shelf thickness. This could amount to a 10% error in PICO melt rates and affect the analysis in non-trivial ways. If the mistake is just a typo, this would be easy to fix. If it is an error in the implementation, the magnitude of the impact will need to be assessed and addressed in the paper. Aside from that one issue, there are a few other small technical issues in the paper that I think can easily be addressed, and which I point out in more detail below.

The manuscript is well written and the figures are clear and well formatted. I have made several suggestions about possible improvements in formatting and grammar below that I think might improve the manuscript. For the most part, I found the organization of the paper to be quite good. The minor exception is that I found that parts of the Discussion section read as if they belong more in the Results, as I will expand on below. And I felt like the Discussion section could dig a bit more into the implications of the choice in the GRISLI implementation of PICO to treat each IMBIE basin as if it were a single ice shelf, rather than

using the original PICO design of treating ice shelves individually. The text mentions that this choice is made for computational efficiency but doesn't go into the potential this has for homogenizing melt patterns across regions with a lot of spatial variability between ice shelves (I'm thinking of the Bellingshausen and Amundsen Seas in particular, where both spatial variability in ocean temperature and in bathymetry depth at calving fronts likely relate to the spatial variability in melting).

My hope is that correcting the calculation of pressure at the ice draft and the reorganization I suggested to the Results and Discussion amount to relatively minor revisions.

**Specific Comments:**

l. 3 and 29 "As a result of limited understanding of these ice-ocean interactions…" and "With our current understanding and computational resources…" My view is that Antarctic ice shelf-ocean interactions are parameterized primarily because of the limited computational resources you point out and also because of difficulties of a more technical (rather than scientific) nature in implementing ice sheet-ocean coupling. A tertiary reason might be the biases in the ocean components of Earth system models that can be corrected as part of applying a parameterization. While I won't deny that there are things we don't understand about ice shelf-ocean interactions, I don't think it is due to a lack of (scientific) understanding that we are using parameterizations. If "limited understanding" was meant to encompass "limited technical understanding", I think it would be important to lean into that a bit more explicitly because I read it as implying "limited scientific understanding" without further clarification.

l. 13 "...we can avoid using ocean temperature bias corrections." That's really great! As I said, above, this is a big advantage of your approach!

l. 69: "$p_k = \rho_{Ice} * g * z_{IceDraft}$": I'm afraid this is my main concern in the paper. This equation need to be either one of these two equivalent forms:

$$p_k = \rho_w \, g \, z_{IceDraft}$$
$$p_k = \rho_I \, g \, h_i$$

where $h_i$ is the full ice thickness of the ice shelf. The form you have here will be too low by a factor of $\rho_i / \rho_w$ or about 0.89 (the value for this ratio given in the text). The error in pressure of about 10% will likely propagate to an error of about 10% in the thermal forcing and therefore the melting from PICO, since the pressure term is a dominant one in the linearization of the freezing point. Please check your implementation to make sure you don't have this error, and correct the text if this is just a typo. If you have implemented this incorrectly, you will need to see what the implications are by rerunning at least a small subset of the calibration with the correction and seeing what the impacts are. This seems large enough that it could change the optimal parameter values.

l. 83: "we decide to make use of the drainage basins defined in (Mouginot and Rignot, 2017) and used in (Rignot et al., 2019)": Please change "make use of" to something more specific. It seems that you treat all ice shelves in each basin as if they were a single ice shelf in PICO.

l. 82-89: Each basin uses the same number of boxes, then, even if it contains some large and some small ice shelves? Did this cause any issues for small shelves with many boxes? I think we found that this gave us trouble in the original PICO and that was the reason for different numbers of boxes for large vs. small ice shelves. I believe we were prone to unrealistic levels of freezing or oscillating thermal forcing between boxes or both. But maybe aggregating the shelves so they act as one makes the difference or perhaps a different calibration is responsible for not seeing this behavior.

l. 105-108: It wasn't clear in this part of the paragraph whether what you describe applies to GRISLI-PICO or PISM-PICO (or both). Could you try to clarify that a bit better?

l. 109: "...outside of the ice shelves in the open ocean…": I don't understand why any ice-sheet model calculation is needed in the open ocean. Shouldn't the growth of ice shelves toward the open ocean be limited by calving, not by parametrized melt? I think more explanation is needed for why you need this additional melt parameterization. It is also unclear where the ocean temperature $T_o$ is being taken from.

l. 120-122: Can you explain how you decided on the upper and lower bounds for these parameters? The results indicate that it's a good range but it would still be helpful to know how you decided what was a reasonable range since too broad a range leads to imprecision in the final calibrated values, while too narrow a range could meant the true optimum is outside the bounds you searched.

l. 124: "...with a 30 years relaxation with GRISLI." Could you say more about how you did the 30-year relaxation with GRISLI? How were the surface and basal fluxes determined, for example?

l. 142-143: "...are up-scaled to the same resolution as the ice sheet model.": So far, you haven't said what that resolution is. Since you say "upscaled", it might be worth stating that that's to either 16 or 40 km resolution.

 Also, what method is used for upscaling? Bilinear interpolation? Conservative remapping?

l. 151 and 153: "pixel-to-pixel": elsewhere in the text, you refer to "points" (e.g. l. 102) or "data points" (e.g. Table 1) rather than "pixels". I think "points" is better than "pixels", since this is not really image data. But I think the best terminology might be "cells", "grid cells" or "model grid cells" to make it clear that the cells of the ice-sheet model grid are what are being referred to in each case.  Here for sure, I would use "cell-to-cell" or (less preferable) "point-to-point" rather than "pixel-to-pixel". Elsewhere, I do think "cell" would be better than "point" or "data point" but I'll leave that up to you.

l. 155-156: I am familiar with these norms being called the Euclidean and Absolute-value error norms, respectively, or also the L-2 and L-1 norms, respectively. The terms (MAE and RMSE) you use are fine, too. But since they are pretty broadly used in statistical and numerical methods, I don't think you need this sentence. The reader can see the difference between them in the table.

l. 156-157: This is great! I appreciate you providing this context here, because it dovetails well with your findings later on that the MAE is a more robust choice than RMSE.

Table 1: "data points": Again, I think it's worth clarifying that these are GRISLI model grid cells, since "data points" is a somewhat vague term.

l. 158-159: "...we do not manage to constrain the method to pick systematically the ensemble members with the best fit to the target.": This phrasing is a little awkward but more importantly I don't think it's clear what you mean by "best fit to the target". It seems like you always find the best fit to the target according to the given metric. I think you need to describe more precisely here what these metrics fail to achieve.

l. 163-164: "...the 20 points with the lowest values…": Should this be "lowest melt-rate values" for clarity?

l. 212-213: "The first of the three, the ADA of bins (panels (g) and (j)), selects almost the same members as the ADA without binning.": It seems to me like the ADA method with and without binning should be (almost) mathematically identical. The average of the cells should be the same as the average over the bins. The only explanation I can come up with for why they might be different is that some bins contain slightly different numbers of grid cells than others, meaning that they cover slightly different areas but still get exactly the same weighting in the binning approach. But can you explain why you think the ADA method with and without binning are not the same? Please include this in the text, not just in your response to my review.

This has implications for your later analysis where you often treat the two ADA results as functionally identical because they choose the same ensemble member. It might be cleaner to just state that the ADA method with binning is so close to the ADA method without binning—any differences are due to poor statistical sampling in each bin—that it isn't worth exploring as a separate approach, and drop it from the rest of the paper.

l. 226: "Nonetheless, being able to match the distribution can also mean spatial compensation between different locations." This spot on, and get at the heart of why many previous calibration methods have not done a great job!

l. 248: "...despite the 169 members of the ensemble…" I think I understand why you bring it up but I do not think the number of ensemble members is a particularly relevant factor here. The only reason the number of ensemble members would play a role is if you had badly undersampled the parameter space and missed a region with a local minimum entirely. The large number of parameter values makes it less likely that you did this, but the difficulty in finding a good set of parameters for BA compared with FRIS is largely for the reasons you get to later in the paragraph. I would leave this phrase out or explain better why it is relevant.

l. 249-250: "This can be explained by the difference in the number of data points.": I doubt very much that this is correct. The smaller number of data points could make the value of the error metric noisier as a function of parameter values but it should not be related to the steep gradient in the error metric with respect to parameter values. That is simply related to

the fact that the melt rates are a much stronger function of the parameters in this region as you point out below. I feel strongly that this sentence should be removed or it should be rephrased to indicate that the smaller number of data points means the error metric will be noisier (but that it is not related to why the metric is a strong function of parameter values).

l. 254-255: Yes! I agree that this is the main reason for the strong gradients in the error metric: at thermal forcing values, the melt rates are much more sensitive to the parameters.

l. 259-261: "The seven cases correspond to the six best members according to the six methods applied Antarctic-wide. It corresponds to five members since the best ADA and best ADA of bins give the same best member." I found this wording quite confusing. Would this be a correct restatement?

> "Five of the seven cases correspond to the best members according to the six methods applied Antarctic-wide. We get five cases rather than six because ADA and ADA of bins give the same best member."

l. 274-275: "The calibration 34, considered as the best one in the calibration process, show very steady ice floating area with a small decreasing trend.": First, I would suggest avoiding subjective words like "very". But the bigger issue here is that the sentence as written seems contradictory. Either the floating area is (very) steady or it has a decreasing trend, and I would say more the latter. How about something like this? "The calibration 34, considered as the best one in the calibration process, shows a small decreasing trend in floating ice area over the course of the simulation compared with most other cases." If you want to say that the trend is small compared to the overall floating area, that would be fine, too. But it should be small with respect to something.

l. 279: It is important to state what the sea level contribution is "low" with respect to. In this case, I think you mean that it is low compared with the other cases you are plotting.

Discussion section: As stated in my general comments, I think this section contains several subsections that would fit better under results. These are 4.1, 4.2 and parts of 4.3 (which part is discussed further below). Each of these sections introduces new calibrations together with error-metric plots similar to those in Section 3. I think these subsections should simply continue section 3 as 3.3, 3.4 and 3.5.

l. 285: When I look at the resolution in plot S57, it looks like Pine Island and Thwaites ice shelves cover only 2 grid cells each (at least in the Paolo data set, not sure about in GRISLI itself). What are the implications of this for PICO? It seems like 2 grid cells for a given ice shelf isn't enough to provide meaningful results? Do you think the parameterization becomes meaningful when you aggregate across basins at this resolution? While I think this subsection should be moved to results, I think these questions could provide the basis for discussion here, after my suggested reorganization.

Figure 8: Why not just include Figure A2 in the text. As far as I can tell, Figure A2 has extra information beyond Figure 8 and there is nothing in Figure 8 that is not in Figure A2. Did you feel that Figure A2 was too complex?

Section 4.3: I would move lines 316 to 326 to the results section for sure. By line 329, you are firmly back in discussion territory. I would probably argue for keeping lines 326 to 329 in the discussion section as well, but that's less clear to me. In any case, this is far beyond my prerogative as a reviewer. I would merely advise that some of the material in this section move to a Results subsection and some remain here as discussion.

l. 326-327 and Figure 10 legends: "The average difference between the two datasets is about 25% of the average value of Adusumilli et al. (2020) at 16 km.": I think the average of the difference is not the right metric to use here. Even if the average of the difference happened to be quite small (and I think that maybe should be the case here), that would not tell you very much about how different the datasets are. Instead, you should use the standard deviation of the difference here. That is a good measure of how different the data sets are from one another. Since that is about triple the mean melt rate (in either data set), it tells you there's a **lot** of disagreement.

l. 329: "...by using the binning methods **we give spatial freedom to the datasets** and constrain them by their values." I think you need to state more clearly what "spatial freedom to the datasets" might mean. I also think it would be good to flesh out the discussion here even more about why you think it's important to get a good statistical representation of melt rates but why the details of the spatial pattern don't matter. More below…

l. 329-331: This is a great argument!

I think it would also be worth pointing out that the Joughin finding isn't a universally accepted result in the field, and instead some researchers (Gudmundsson 2013, doi:10.5194/tc-7-647-2013; Reese et al. 2017 https://doi.org/10.1038/s41558-017-0020-x for two) think that buttressing plays an important role and melt in more strongly buttressed areas has an outsized effect.

Figure 10: In the caption, could you make clear that these results are for the 16 km GRISLI mesh? It wasn't clear to me until I looked at the supplementary material.

Also, could you help me understand why the difference between the means in panel a would be about 0.02 m/yr but the mean of the difference is 0.16 m/yr? My guess is that there are many cells where only one of the datasets has data, and that accounts for the difference but wanted to make sure. Otherwise, I would have expected the difference of the means to be the mean of the differences.

Section 4.4: This is the first section within the discussion that I think fully belongs here. It is full of some really great points.

l. 350-362: Be careful that you present these differences in a neutral tone. You can't know for sure whether the PICO or the quadratic nonlinear results are more accurate since we don't know the right answer for future response. The only room there is to criticize one parameterization in relation to another is based on the physics each includes or excludes.

l. 350-351: "PICO includes overturning circulation under the ice shelves, which tends to reduce the basal melt rate. This freshwater negative feedback is not included in the QuadNL parameterisation." I think this is a misunderstanding of the quadratic parameterization. Like PICO, the quadratic parameterization has one term involving the thermal forcing that is meant to represent how warmer temperatures near the ice-ocean interface lead to higher melt rates. But the second term in the product involving the thermal forcing is meant to represent the stronger overturning that results from warmer conditions. This is essentially equivalent to the PICO term involving the overturning coefficient C. See, for example, Holland et al. (2008, doi:10.1175/2007JCLI1909.1), Jourdain et al. (2017, doi:10.1002/2016JC012509), Burgard et al. (2022, doi:10.5194/tc-2022-32). So I do not agree with the statement on this line and think it needs to be heavily revised or removed.

l. 352-359: These are 3 excellent points!

l. 360-362: I really like the point about sensitivity to oceanic forcing in QuadNL. But I take issue with "therefore this QuadNL calibration could overestimate basal melt rates". I don't think this follows. The lack of refreezing certainly will mean that, if the method is calibrated to get the right mean melt, it will also underestimate melt maxima. But it doesn't follow that it will overestimate melt on the whole, this depends on the detail of the calibration approach. I would remove this phrase.

l. 363-364: I would think this would be fairly obvious to anybody doing ice sheet modeling. There is not necessarily any relationship between the area of floating ice (the ice shelves) and grounded ice, which is the only part that contributes to sea level. Even the relationship between the grounded area and the volume (or mass) above flotation can be complicated. But I have not generally seen the area of floating ice presented as a quantity of major dynamical significance for ice sheet models in the past. (But I will add that ice-sheet modeling is less my field.)

l. 364-366: I'm afraid I don't quite follow this argument. It's not obvious to me why different time series of ice-shelf area would imply different pathways. Ice shelf area could be reduced along the same pathway just by the presence of more (or less) calving relative to grounding-line motion.

l. 368-369: "This methodology can potentially be applied to modules in other models that benefit from existing observational datasets." This is an excellent point! I think this work has exciting potential applications to other methods and models.

l. 379: "...spatial contrast…": I find this phrase a little inexact. How about "range of special variability, if not the details of spatial patterns"?

l. 390-391: This is an excellent point! The paleo record is one good way to constrain this work over longer times. Perhaps high fidelity coupled ice sheet-ocean simulations will provide another such method in the future.

l. 398-399: I appreciate you providing this recommendation! I think many modelers will find it useful.

l. 401-402: "But also, as the present study has the limitation of targeting a given basal melt rate for a given ocean temperature rather than the sensitivity of the basal melt rate to changes of ocean temperatures." I think this is a great point but it isn't clear how it follows from the previous sentence. How will more modelers testing the calibration method address this limitation?

l. 404-406: "Alternatively, the low sensitivity of PICO in comparison to QuadNL could also be adjusted by developing a quadratic dependence to thermal forcing that would give a quadratic parameterisation that also accounts for overturning circulation under the ice shelves." Presumably, it will come as no surprise given my statements above that I do not agree with this statement. I believe the QuadNL parameterization does already take the overturning into account.

**Typographical, Grammatical and Formatting Suggestions:**

l. 22: "Sub-surface melt of the ice shelves **on the other hand**…" It's not clear what "on the other hand" is contrasting here. Maybe something else like "in turn" is more appropriate?

l. 39: "...distribution of values from the observations **as best as possible.**" The phrase "as best as possible" is often used informally but probably should be replaced with something like "to the extent possible" in scientific writing.

l. 45: "...whether it matters to calibrate PICO at a smaller scale than Antarctic wide…" This took me a little time to understand. Maybe rephrase this for clarity as something like "...whether regional calibrations of PICO produce different results than those applied over all of Antarctica…"

l. 65 and throughout the subsequent text and figures: "Sv.m$^3$.kg$^{-1}$": You consistently use periods to separate units. I imagine this is something the typesetter will handle but my experience with TC is that they use half-spaces (`\,` in latex) between symbols. I have seen a multiplication dot (`\cdot` in latex) used occasionally in other journals but never a period. I would advise changing to a half space.

l. 68: "under-burden pressure": This may just relate to our difference in point of view but I had not heard of this term used in this context before. I guess from an ice-sheet modeling perspective, this is the pressure of the ocean pushing up on the ice? From my ocean modeling perspective, this is the overburden pressure of the weight of the ice pressing down on the ocean. Maybe just call it the "pressure at the ice-ocean interface" or something similar to be agnostic to these two perspectives?

l. 69: "$p_k = \rho_{Ice} * g * z_{IceDraft}$": I would suggest using multiplication dots (`\cdot` in latex) here or just spaces but "*" is not the right thing to use in scientific writing. For consistency with what you have on l. 74, $\rho_{Ice}$ should be $\rho_I$.

l. 75-77: "The coefficient a is the salinity coefficient of the freezing equation…pressure coefficient of the freezing equation and equals $7.77 \times 10^{-8}$ °C Pa$^{-1}$." This simplified equation is typically referred to as a linearization of the "equation of state for the freezing point of

seawater". This might be better than "freezing equation." The coefficients are often called the liquidus slope, liquidus intercept and liquidus pressure coefficient, respectively (see, for example, Asay-Davis et al. (2016, doi:10.5194/gmd-9-2471-2016) – no need to cite this, just pointing to the terminology there).

l. 83: "...we decide to make use of the drainage basins defined in (Mouginot and Rignot, 2017) and used in (Rignot et al., 2019)": These citations should be outside of parentheses (`\citet` instead of `\citet` in latex).

l. 95: "All the four criteria are here not followed…": This would be significantly clearer as something like, "Here, we do not follow any of these criteria…"

l. 98: "...show in some areas refreezing…" should be "...show refreezing in some areas…"

l. 102-103: "...we consider as grounding line any ice points that is surrounded by grounded ice and not grounded ice…": I found this a little hard to follow and I think it would be clearer as something along the lines of, "...we consider any ice points to be on the grounding line if it has some neighbors that are grounded and others that are floating…"

l 108, 113, 210: "...enables to have…" and "...enables to limit..." should be something like "...enables us to have…" or "...enables one to have…" (less preferable). This particular passive form is not grammatically correct in English.

l. 124: "...30 years relaxation…" should be "...30-year relaxation…"

l. 146 and 292; "We present first the three ones…" and "...the methods RMSE of bins and MAE of bins are the two ones..": It would be more appropriate for scientific writing to replace "three ones" and "two ones" with "three methods" and "two methods", respectively. (I know it sounds redundant, but there doesn't seem to be a good way to avoid that.)

l. 151 and 153: "two Dimensions…" should be "two-dimensional"

l. 159: "...we decide to proceed a binning on the dataset…": I'm not entirely sure what you are saying here but maybe, "...we decide to bin each dataset…"

l. 160 "method" should be "the method"

l. 161: Here and later in the text, I believe the standard is to use a capital F in "Figure" (and T in "Table") even within a sentence when referring to figures by number. I think you do this later in the text in a few places, but mostly you use lowercase f, and I think you should use capital F throughout.

l. 176-177: "For that…" isn't super clear or standard grammar. How about, "to accomplish this…"

l. 183: "...in line with…": This is super picky but I think this should just be "...using…" because "in line with" implies that the results are consistent with what you're about to refer to, not that they were produced using what you're about to refer to.

l. 198, 200, 266, 271, 291 and elsewhere: "on panel" or "on figure" should always be "in panel" or "in figure".

l. 207: "...will not be systematically in the same order of magnitude of values…" might be easier to follow as something like "...will not systematically be of the same order of magnitude…"

Figure 3: I would put the panel letters before the text they refer to, not after: "...to each methods, (a) ADA of bins, (b) RMSE of bins and (c) MAE of bins…" This is more standard notation.

l. 229: "...they even do not have values…" should be "...they do not even have values…"

l. 235: "...the possibility to compensate…" should be "...the possibility that there is compensation…"

l. 279: "result all" should be "all result"

l. 283: "...discuss further about the ISMIP 2300 results obtained using PICO" should be "...discuss the ISMIP 2300 results obtained with PICO further."

l. 290, 291, 313: "cumulating": Although "to cumulate" appears to be a valid English verb, as a native speaker I had never heard of it before. I would suggest using "accumulate," "aggregate," or "combine" instead.

l. 294-296: This could be cleaned up a little as: "This consistency can matter to inter-compare: i) different parameterisations ; ii) when the same parameterisation **is** used in different ice sheet models ; iii) results at different resolutions." All 3 use "inter-compare" so it can be brought out front and "is" is missing.

l. 306: An end parenthesis ")" is missing before the end of the sentence.

l. 307: "...we present in figure 9 results…" should be "...in Figure 9, we present results…"

l. 318: "Hence, we could argue to choose a different target…" would be more grammatical as something like , "Hence, we could make an argument for choosing a different target…"

l. 321: "They are overall similar.." should be, "Overall, they are similar…"

l. 328: "...important to justify…" should be "...important in justifying…"

l. 331: "...the resolution **of** the observational datasets…"

l. 334-335: "...also consider to compute…" should be "...also consider computing…"

l. 333-335: These are also great points of discussion.

l. 345: I would suggest: "Elmer-ice with PICO **even suggests** a negative sea level contribution from Antarctica **throughout** the simulation."

l. 347-348: "...with PICO as shown with the simulation with PISM." That's a lot of "withs". I'd suggest, "...with PICO as shown **in** the simulation with PISM."

l. 348: "But **here** we want to provide…"

l. 358: "...does not peak that high at grounding lines": again I take issue with the qualitative and subjective nature of this phrase. How about something like "...does not show significantly higher melt rates at grounding lines, as seen in satellite-derived fields"?

l. 367: "fits best" should be "best fits"

l. 377: "...forces to fit the target values also at the low and high extremes…" should be something like "...that also fit the low and high extremes in the target histogram…"

l. 378: "systematic" should be "consistent"

l. 380: "...matches well the entire distribution of…" could be something like "...does a good job at matching the entire distribution…"

l. 388-389: "ice shelves extent" should be "ice-shelf extent" and "ice shelves changes" should be "ice-shelf changes"

l. 394-395: "...could avoid modellers to use temperature corrections…": How about this? "...could avoid the need for modelers to use temperature corrections…"

---

## Referee Comment (RC2)

Review for Menthon et al. : **Comparison of calibration methods of a PICO basal ice shelf melt module implemented in the GRISLI v2.0 ice sheet model**

Submitted to *Geoscientific Model Development*

*Reviewer: Clara Burgard*
*I do this review un-anonymously to clarify from which background and level of expertise the comments come from and because it makes the conversation during review more transparent on both sides.*

**Summary**

Ocean-induced melt of Antarctic ice shelves and its representation in ice-sheet models is one of the main sources of uncertainty for ice-sheet projections. The parameterisations used to bridge the gap between ocean properties and the basal melt remain uncertain. The authors revisit this uncertainty and explore a new calibration approach to reduce it. To do so, they implement the PICO ice-shelf basal melt parameterisation into the ice-sheet model GRISLI and test different novel calibration approaches, resulting in a more robust parameter choice. They explore a variety of conditions that affect the calibration such as the metric, the resolution, the geographical specificities, and the forcing conditions.

The study is a timely contribution because the parameterisation of the melt at the base of Antarctic ice shelves remains one of the largest sources of uncertainty in projections of the future evolution of the Antarctic ice sheet and its contribution to sea-level rise. The approach explored by the authors is a useful addition to the community as it provides a set of parameters that seem to be more robust across conditions compared to previous sets. It could also be explored for other parameterisations in the future.

The manuscript is very clear and thorough. This thorough description is very well suited to inspire other researchers to explore the presented methods on other parameterisations and/or other ice-sheet models. Overall, I have a few larger comments as some aspects of the study appeared unclear to me and other more minor comments. I therefore suggest minor revisions.

**GENERAL COMMENTS**

I want to start by thanking the authors for making this manuscript very pleasant to as the manuscript and figures are very well done and clear.

**#1** The results of this study are very encouraging. However, there is a strong limitation, which is not discussed much. In Figure 3, it becomes clear that using bins improves the melt rates for the "middle" bins but, in all cases, the spread between members remains high for the low and high-end bins. In particular, for the lowest melt, an anomaly of 0.5 m per yr could be quite high compared to the actual melt. For the highest melt, the order of magnitude is maybe lower compared to the actual melt. However, it is important to check this further because the points with the highest melt are also the points that influence the future of the ice sheet the most. Would it be possible to complete this by a metric that looks at the percentage formed by this anomaly compared to the melt value? Just to have a better idea of what this means exactly.

**#2** I am not completely convinced by section 4.2. I do not understand how the authors can calibrate on different input temperatures (e.g. 1K warmer) but same target melt and make conclusions from this. If the parameters do not change much, does that not mean that they are not sensitive enough to temperature changes? Ideally, you would expect higher melt for higher temperatures if you use the same set of parameters, no? This makes me unsure about the whole approach. Does that mean that the parameterisation cannot react to changes in forcing and that the parameters are too strongly set? This would not be useful for projections and is what could be interpreted from the low sea-level rise contribution in Fig. 7c. To reassure the reader, I recommend that the authors clarify the implications of this result, discuss them more in detail or reformulate to avoid misunderstanding.

**DETAILED CONTENT COMMENTS**

**L3 :** It is not only about limited computational resources but also about a lack of observational data. I suggest completing "limited observational data and computational resources".

**L31:** Not sure if the formulations only differ in the complexity of the "melt physics" themselves. In most cases, assumptions differ on the simplification of the ocean circulation in the cavity. Can the authors reformulate to clarify?

**L39-40:** This is great! Calibrations that do not need regional corrections are what are currently needed!

**L104-108:** This paragraph is unclear in regard to the authors' own contribution vs what has been done in PICO-PISM before. I suggest reformulating it to clarify.

**L114:** It is unclear where the value for the combined factor comes from. As far as I know $K\_T$ is a parameter that needs calibration. So how has it been calibrated in this case? I suggest the authors add the source or briefly explain the calibration.

**L122-124:** It remains unclear to me why the authors use the coupled GRISLI-PICO setup if the geometry is kept constant anyway. Could the authors clarify why they do not use a standalone version of PICO then?

**Table 1:** Very useful table!

**Table 1:** Maybe I did not think this completely through but aren't ADA and ADA of bins the same metric. Taking the mean of a mean should, I think, result in the same as taking the mean of the whole ensemble directly. This would also explain why the results are so similar between the two. Can the authors comment on that?

**L218-219:** I am not 100% sure that a narrow set of parameters is a guarantee for "better". I agree it is useful for modellers but I would be careful with such kind of statements. If a large range of parameters are possible, this could also be linked to the formulation of the

parameterisation. Still, in L294-296, the authors explain the advantages of having a narrow set of parameters. Maybe this could already be mentioned here?

**L241:** It is unclear to me what is the difference between the distribution curve and the magnitude of the spatial patterns. Is one the shape of the curve and the other the actual number?

**Figure 8:** Just a remark: The 2D results are interesting and give the feeling that C does not really play a role in the calibration. Out of curiosity, have the authors thought about what this could mean?

**L330:** Agreed that Joughin et al. (2021) is a good study to refer to here but it should not be forgotten that enough other studies (e.g. Reese et al. 2018) show that localised melt has a strong effect on buttressing. I suggest that the authors reformulate a little more carefully.

**L350:** This is not completely true. The quadratic term of the quadratic parameterisation is there to mimic the effect of the overturning circulation in a very simple way.

**L360-362:** Yes, it can be clearly seen in the sensitivities to warmer forcing in Burgard et al. (2023) and Lambert and Burgard (in press) that the quadratic parameterisation is an outlier towards high melt sensitivity. However, as we do not know what is the "right" sensitivity, this is not enough to say that one is better than another.

**L363-366:** This is not very clear. I suggest that the authors reformulate to clarify.

**DETAILED WRITING COMMENTS**

**L20:** Replace "warmth" by "heat"

**L22:** "on the other hand" does not really work in this sentence. I suggest leaving it out.

**L34:** Leave out "However", it is confusing.

**L41-49:** This could be shortened.

**L54:** Replace "are dependent" by "depend"

**L79 and 84:** For the results, I suggest to stay consistent with present instead of past tense: "made" => "make", "combined" => "combine"

**L83:** Correct the citation format (\citet{} instead of \citep{})

**L98:** Rephrase to "refreezing in some areas"

**L142:** Replace "but also" by "and"

**L148:** To improve reproducibility, I suggest that the authors add the information if the results are in m ice per year or in m w.e. per year.

**L158:** I suggest that the authors reformulate, the formulation is very unclear.

**Figure 1 caption, last sentence:** Replace "is" by "are"

**L190:** Can the authors clarify if they are writing about the sensitivities of PICO and QuadNL to ocean warming or to something else?

**L209 and later:** "side-by-side" sounds awkward. I suggest reformulating, maybe with something like "close" or "similar".

**L220:** Replace "explain" by "explained"

**L224-225:** I suggest reformulating as this is not a complete sentence.

**L232:** "led" => "lead"

**L235:** remove "is"

**L236:** missing "methods" in front of "without"

**L269:** "Elemer" => "Elmer"

**Figure 6 and later:** Would it be possible to replace CalibXX by an indication of the metric it was calibrated with? That would clarify the legend.

**L284:** I suggest reformulating "we further discuss the ISMIP 2300 …"

**L359:** "rates" => "rate"

**L401-402:** This is not a complete sentence.

**Supplementary material:** There is A LOT of material and the captions are sometimes very short. I wonder if it would be possible to reduce the amount of figures or add one sentence explaining the core of the figure or set of figures when appropriate?

*References*
- Reese et al. 2018 : https://doi.org/10.1038/s41558-017-0020-x
- Burgard et al. 2023: https://doi.org/10.1029/2023MS003829
- Lambert and Burgard, in press: https://doi.org/10.5194/egusphere-2024-2358

---

## Author Comment (AC1)

**Review of Menthon et al. "Comparison of calibration methods of a PICO basal ice shelf melt module implemented in the GRISLI v2.0 ice sheet model"**

Reviewer: Xylar Asay-Davis

I wish my name to be relayed to the authors, as I feel I am always a better reviewer when I am not anonymous and I encourage others to consider reviewing non-anonymously whenever they feel able.

> We would like to thank the reviewer, Xylar Asay-Davis, for taking the time to review the paper by providing detailed comments to improve the manuscript. We address all the points raised below, our responses to the comments are in blue.

**General Comments:**

This paper presents the implementation of the PICO parameterization for Antarctic basal melting in the GRISLI v2.0 ice sheet model. Then, the authors analyze a series of potential methods for parameter calibration, settling on a binning technique together with the mean absolute error (MAE) as the most robust approach. They probe the robustness of the approach in several ways, including modifying the ocean forcing, changing melt-rate dataset use as the observational target, calibrating regionally rather than Antarctica-wide, and using different ice-sheet resolutions (also the resolution of the parameterization). A significant finding is that their calibration method requires no correction of the input ocean temperature field, in contrast to previous work. In addition to the calibration, they ran an ensemble of projection simulations to the year 2300, showing the impacts on total melt flux, floating ice area, and Antarctic mass change from different PICO calibrations.

The paper is quite detailed and for a fairly niche audience. Even so, I think the level of detail is justified because the analysis performed here is potentially applicable not just to the PICO parameterization but to all Antarctic basal melt parameterizations. The results present a strong case that the binning technique used for calibration leads to parameter choices that are robust to the various uncertain inputs to the parameterization and which represent the statistics if not the detailed spatial patterns of Antarctic basal melting.

I see a potentially significant issue in the equation presented for the pressure at the ice shelf-ocean interface that could impact the results. As I expand on in my specific comments below, the pressure as presented is computed from the ice density but the ice draft (the ocean thickness, so to speak), whereas the correct pressure must involve either the ocean density and the ice draft or the ice density and full ice-shelf thickness. This could amount to a 10% error in PICO melt rates and affect the analysis in non-trivial ways. If the mistake is just a typo, this would be easy to fix. If it is an error in the implementation, the magnitude of the impact will need to be assessed and addressed in the paper. Aside from that one issue, there are a few other small technical issues in

the paper that I think can easily be addressed, and which I point out in more detail below.

The manuscript is well written and the figures are clear and well formatted. I have made several suggestions about possible improvements in formatting and grammar below that I think might improve the manuscript. For the most part, I found the organization of the paper to be quite good. The minor exception is that I found that parts of the Discussion section read as if they belong more in the Results, as I will expand on below. And I felt like the Discussion section could dig a bit more into the implications of the choice in the GRISLI implementation of PICO to treat each IMBIE basin as if it were a single ice shelf, rather than using the original PICO design of treating ice shelves individually. The text mentions that this choice is made for computational efficiency but doesn't go into the potential this has for homogenizing melt patterns across regions with a lot of spatial variability between ice shelves (I'm thinking of the Bellingshausen and Amundsen Seas in particular, where both spatial variability in ocean temperature and in bathymetry depth at calving fronts likely relate to the spatial variability in melting).

My hope is that correcting the calculation of pressure at the ice draft and the reorganization I suggested to the Results and Discussion amount to relatively minor revisions.

**Specific Comments:**

l. 3 and 29 "As a result of limited understanding of these ice-ocean interactions…" and "With our current understanding and computational resources…" My view is that Antarctic ice shelf-ocean interactions are parameterized primarily because of the limited computational resources you point out and also because of difficulties of a more technical (rather than scientific) nature in implementing ice sheet-ocean coupling. A tertiary reason might be the biases in the ocean components of Earth system models that can be corrected as part of applying a parameterization. While I won't deny that there are things we don't understand about ice shelf-ocean interactions, I don't think it is due to a lack of (scientific) understanding that we are using parameterizations. If "limited understanding" was meant to encompass "limited technical understanding", I think it would be important to lean into that a bit more explicitly because I read it as implying "limited scientific understanding" without further clarification.

> Thanks for raising this, we clarified with the following text:
>
> "As a result of technical challenges related to computational resources, implementation and different modelling time-scales, these interactions are often parameterised rather than explicit resolved in ice sheet models"

l. 13 "...we can avoid using ocean temperature bias corrections." That's really great! As I said, above, this is a big advantage of your approach!

> Thanks!

l. 69: "pk = ρIce ∗ g ∗ zIceDraft": I'm afraid this is my main concern in the paper. This equation need to be either one of these two equivalent forms:

pk = ρw g zIceDraft

pk = ρI g hi

where hi is the full ice thickness of the ice shelf. The form you have here will be too low by a factor of $\rho_i / \rho_w$ or about 0.89 (the value for this ratio given in the text). The error in pressure of about 10% will likely propagate to an error of about 10% in the thermal forcing and therefore the melting from PICO, since the pressure term is a dominant one in the linearization of the freezing point. Please check your implementation to make sure you don't have this error, and correct the text if this is just a typo. If you have implemented this incorrectly, you will need to see what the implications are by rerunning at least a small subset of the calibration with the correction and seeing what the impacts are. This seems large enough that it could change the optimal parameter values.

> Thanks for pointing this out! It was a typo, and we corrected this, the text reads as follow:
>
> [...] the ice draft to calculate the under-burden pressure under the ice shelf pk using pk = ρSeaWater ∗ g ∗ zIceDraft, with ρSeaWater = 1033 kg.m−3.

l. 83: "we decide to make use of the drainage basins defined in (Mouginot and Rignot, 2017) and used in (Rignot et al., 2019)": Please change "make use of" to something more specific. It seems that you treat all ice shelves in each basin as if they were a single ice shelf in PICO.

> We changed this to: "To reduce computational costs we avoid identifying individual ice shelves in each time step, instead we decide to solve one instance of the PICO equations per drainage basin, with the basins as presented in Mouginot and Rignot (2017) and used in Rignot et al. (2019), ..."
>
> This is indeed the case, all ice shelves in one drainage basin are seen as one even though they might be physically separated. To clarify, we added the following at the end of the paragraph:
>
> "This simplification of solving one PICO instance per drainage basin enables us to compute faster for each ice shelves their number of boxes, as well as their corresponding temperature and salinity inputs."

l. 82-89: Each basin uses the same number of boxes, then, even if it contains some large and some small ice shelves? Did this cause any issues for small shelves with many boxes? I think we found that this gave us trouble in the original PICO and that was the reason for different numbers of boxes for large vs. small ice shelves. I believe we were prone to unrealistic levels of freezing or oscillating thermal forcing between boxes or both. But maybe aggregating the shelves so they act as one makes the difference or perhaps a different calibration is responsible for not seeing this behavior.

Yes, in each basin, we have a fixed number of boxes, while they can differ in-between basins. We found that this is not too much of an issue, as the overall size of ice shelves in each basin is comparable, see SI Figure 1. We have added the following explanation:

"The number of boxes in each drainage remains relative to the maximum distance between the ice shelf front and the grounding line of all the ice shelves, as defined in the equation 1. And since within the same drainage basin there are roughly similar sizes of ice shelves, the division by drainage basin do not cause discrepancies such as small ice shelves with up to five boxes or larger ice shelves with few boxes (see Supplement Figure 1)."

[Figure]

l. 105-108: It wasn't clear in this part of the paragraph whether what you describe applies to GRISLI-PICO or PISM-PICO (or both). Could you try to clarify that a bit better?

Clarification were made as follow, with the new parts in bold: **"The grounding line can be defined in different ways and therefore can lead to different PICO boxes geometries. Here, in GRISLI-PICO,** we consider as grounding line any ice points that is surrounded by grounded ice and not grounded ice, and as ice front any ice point that is floating and adjacent to ocean. **Whereas, in** PISM-PICO (Reese et al., 2018) they did not include the grounding line of ice rises and also excluded holes in ice shelves as ice-shelf front when identifying PICO boxes. The grounding lines of ice rises are defined as not being directly connected to the main grounded part of the ice sheet which is identified by the size of the connected grounded region. Thus, **in PISM-PICO i**t is possible to have ice shelves without grounding line connected to the main ice sheet, where PICO cannot define a box geometry. In these places, the parametrisation of Beckmann and Goosse (2002) **was used in PISM** to have a rough estimate of the basal melt rates."

l. 109: "…outside of the ice shelves in the open ocean…": I don't understand why any ice-sheet model calculation is needed in the open ocean. Shouldn't the growth of ice shelves toward the open ocean be limited by calving, not by parametrized melt? I think more explanation is needed for why you need this additional melt parameterization. It is also unclear where the ocean temperature To is being taken from.

This is very ice-sheet model dependant I think. In our case, GRISLI needs to know the mass balance and the lagrangian ice flux to know whether the ice shelf front it advancing towards the ocean or not, therefore we need to provide an estimate of the basal melt rate also where there is currently no ice. To clarify this point we added and reformulated the following text:

"GRISLI incorporates a dynamic calving front that advances based on a balance between the Lagrangian ice flux and local surface and basal mass balances \citep{Quiquet2018-gmd}. To evaluate potential ice shelf advance at each timestep, the model must compute these mass balances even beyond the current ice extent. Unlike alternative approaches such as the level set method, which do not require mass balance information outside the ice mask, this is a necessary feature for GRISLI. Thus, in regions beyond the ice shelf, over open ocean, we apply the parameterisation of \cite{DeConto2016-nature}, defined as follows:"

To clarify about To we added the following:

"The temperature input $T_0$ is computed the same way as for PICO."

l. 120-122: Can you explain how you decided on the upper and lower bounds for these parameters? The results indicate that it's a good range but it would still be helpful to

know how you decided what was a reasonable range since too broad a range leads to imprecision in the final calibrated values, while too narrow a range could meant the true optimum is outside the bounds you searched.

> We added the following to justify our range choice: "The range of values for the two parameters has been chosen based on literature (Reese et al., 2018; Burgard et al., 2022; Reese et al., 2023) and adjustments in such a way that the best values are not on one of the extremes of the range of values."

l. 124: "...with a 30 years relaxation with GRISLI." Could you say more about how you did the 30-year relaxation with GRISLI? How were the surface and basal fluxes determined, for example?

> During the basal drag coefficient inversion methodology used for ISMIP6, we compute the ice sheet internal thermal equilibrium with a long (60 kyr) experiment with fixed observed geometry. To avoid any artificial drift when releasing this constraint we first run a 30 yr relaxation experiment with the same boundary conditions as for the control experiment (ctrl in ISMIP6).

l. 142-143: "...are up-scaled to the same resolution as the ice sheet model.": So far, you haven't said what that resolution is. Since you say "upscaled", it might be worth stating that that's to either 16 or 40 km resolution.

Also, what method is used for upscaling? Bilinear interpolation? Conservative remapping?

> We added the following to clarify: " ... are up-scaled to the same resolution as the ice sheet model **(16 km or 40 km), using CDO bilinear interpolation."**

l. 151 and 153: "pixel-to-pixel": elsewhere in the text, you refer to "points" (e.g. l. 102) or "data points" (e.g. Table 1) rather than "pixels". I think "points" is better than "pixels", since this is not really image data. But I think the best terminology might be **"cells", "grid cells" or "model grid cells"** to make it clear that the cells of the ice-sheet model grid are what are being referred to in each case. Here for sure, I would use **"cell-to-cell"** or (less preferable) "point-to-point" rather than "pixel-to-pixel". Elsewhere, I do think "cell" would be better than "point" or "data point" but I'll leave that up to you.

> This has been corrected in the whole paper.

l. 155-156: I am familiar with these norms being called the Euclidean and Absolute-value error norms, respectively, or also the L-2 and L-1 norms, respectively. The terms (MAE and RMSE) you use are fine, too. But since they are pretty broadly used in statistical and numerical methods, I don't think you need this sentence. The reader can see the difference between them in the table.

The first sentence has been removed as suggested. And the second sentence has been added to the description of the 2D MAE which now reads as follow:

Two Dimensions Mean Absolute Error (2D MAE): we compute the MAE cell-to-cell with the same geographical location between each ensemble member and the target. **Since no squaring is used in the error computation in the MAE makes the MAE less sensitive to outliers than the RMSE.**

l. 156-157: This is great! I appreciate you providing this context here, because it dovetails well with your findings later on that the MAE is a more robust choice than RMSE.

This is kept and moved as mentioned in the reply to the previous comment.

Table 1: "data points": Again, I think it's worth clarifying that these are GRISLI model grid cells, since "data points" is a somewhat vague term.

All "data points" have been changed to "grid cells".

l. 158-159: "…we do not manage to constrain the method to pick systematically the ensemble members with the best fit to the target.": This phrasing is a little awkward but more importantly I don't think it's clear what you mean by "best fit to the target". It seems like you always find the best fit to the target according to the given metric. I think you need to describe more precisely here what these metrics fail to achieve.

We changed the text as follow to clarify it: "By applying the three first ranking methods, the ranking metrics do not enable to pick systematically the ensemble members with the best fit to the distribution of values of the observational dataset."

l. 163-164: "…the 20 points with the lowest values…": Should this be "lowest melt-rate values" for clarity?

Adjusted as suggested.

l. 212-213: "The first of the three, the ADA of bins (panels (g) and (j)), selects almost the same members as the ADA without binning.": It seems to me like the ADA method with and without binning should be (almost) mathematically identical. The average of the cells should be the same as the average over the bins. The only explanation I can come up with for why they might be different is that some bins contain slightly different numbers of grid cells than others, meaning that they cover slightly different areas but still get exactly the same weighting in the binning approach. But can you explain why you think the ADA method with and without binning are not the same? Please include this in the text, not just in your response to my review.

This has implications for your later analysis where you often treat the two ADA results as functionally identical because they choose the same ensemble member. It might be

cleaner to just state that the ADA method with binning is so close to the ADA method without binning—any differences are due to poor statistical sampling in each bin—that it isn't worth exploring as a separate approach, and drop it from the rest of the paper.

> We agree with your point and explanation to the very small differences that we observe, we therefore added the following in the paper:
>
> "This method do not leads to exactly the same results as the ADA without binning because some bins might contain a different numbers of grid cells than others, meaning that they cover slightly different areas but still get exactly the same weighting in the binning approach. As we intend to compare methodologies, we consider it relevant to also test this method, even if it gives results very close to ADA without binning, to quantify how much they differ."

l. 226: "Nonetheless, being able to match the distribution can also mean spatial compensation between different locations." This spot on, and get at the heart of why many previous calibration methods have not done a great job!

> Thank you.

l. 248: "…despite the 169 members of the ensemble…" I think I understand why you bring it up but I do not think the number of ensemble members is a particularly relevant factor here. The only reason the number of ensemble members would play a role is if you had badly undersampled the parameter space and missed a region with a local minimum entirely. The large number of parameter values makes it less likely that you did this, but the difficulty in finding a good set of parameters for BA compared with FRIS is largely for the reasons you get to later in the paragraph. I would leave this phrase out or explain better why it is relevant.

> We removed "…despite the 169 members of the ensemble…" from the text.

l. 249-250: "This can be explained by the difference in the number of data points.": I doubt very much that this is correct. The smaller number of data points could make the value of the error metric noisier as a function of parameter values but it should not be related to the steep gradient in the error metric with respect to parameter values. That is simply related to

the fact that the melt rates are a much stronger function of the parameters in this region as you point out below. I feel strongly that this sentence should be removed or it should be rephrased to indicate that the smaller number of data points means the error metric will be noisier (but that it is not related to why the metric is a strong function of parameter values).

> We agree and removed "This can be explained by the difference in the number of data points." from the text.

l. 254-255: Yes! I agree that this is the main reason for the strong gradients in the error metric: at thermal forcing values, the melt rates are much more sensitive to the parameters.

Thank you.

l. 259-261: "The seven cases correspond to the six best members according to the six methods applied Antarctic-wide. It corresponds to five members since the best ADA and best ADA of bins give the same best member." I found this wording quite confusing. Would this be a correct restatement?

"Five of the seven cases correspond to the best members according to the six methods applied Antarctic-wide. We get five cases rather than six because ADA and ADA of bins give the same best member."

Indeed, your suggestion is clearer, we changed the text as suggested.

l. 274-275: "The calibration 34, considered as the best one in the calibration process, show very steady ice floating area with a small decreasing trend.": First, I would suggest avoiding subjective words like "very". But the bigger issue here is that the sentence as written seems contradictory. Either the floating area is (very) steady or it has a decreasing trend, and I would say more the latter. How about something like this? "The calibration 34, considered as the best one in the calibration process, shows a small decreasing trend in floating ice area over the course of the simulation compared with most other cases." If you want to say that the trend is small compared to the overall floating area, that would be fine, too. But it should be small with respect to something.

We agree and changed the text as suggested.

l. 279: It is important to state what the sea level contribution is "low" with respect to. In this case, I think you mean that it is low compared with the other cases you are plotting.

We clarified as follow: "Except compared to Elmer-ice with PICO, all the simulations of GRISLI-PICO result in lower values of sea level contribution by 2300 than all the other cases."

Discussion section: As stated in my general comments, I think this section contains several subsections that would fit better under results. These are 4.1, 4.2 and parts of 4.3 (which part is discussed further below). Each of these sections introduces new calibrations together with error-metric plots similar to those in Section 3. I think these subsections should simply continue section 3 as 3.3, 3.4 and 3.5.

We implemented the suggested reorganization.

l. 285: When I look at the resolution in plot S57, it looks like Pine Island and Thwaites ice shelves cover only 2 grid cells each (at least in the Paolo data set, not sure about in GRISLI itself). What are the implications of this for PICO? It seems like 2 grid cells for a

given ice shelf isn't enough to provide meaningful results? Do you think the parameterization becomes meaningful when you aggregate across basins at this resolution? While I think this subsection should be moved to results, I think these questions could provide the basis for discussion here, after my suggested reorganization.

> This is indeed an important limitation. Yet, it is challenging to provide another better alternative for models using this resolution. We consider the results obtained even for few grid-cells ice shelves to be reasonably good.
>
> We integrated the following in the text:
>
> "Consequently, certain ice shelves in the West Antarctic ice sheet such as Thwaites or Pine Island have very few grid cells at 40 km resolution. This can be an important limitation, yet we consider important to test it and potentially provide an option for paleo ice sheet simulations."

Figure 8: Why not just include Figure A2 in the text. As far as I can tell, Figure A2 has extra information beyond Figure 8 and there is nothing in Figure 8 that is not in Figure A2. Did you feel that Figure A2 was too complex?

> This is right Figure A2 has more information than Figure 2. We made it this way in order to facilitate the attention of the reader to the point we are making: how big is the range of best values for the two parameters in all the six conditions. Adding the resolution is making the figure less clear, and would discontinue a bit the type of figure done all along the paper (such as Figures 2, 9, 11). However, the reader will likely ask whether there are patterns due to resolution, therefore we consider important to add it in the appendix.

Section 4.3: I would move lines 316 to 326 to the results section for sure. By line 329, you are firmly back in discussion territory. I would probably argue for keeping lines 326 to 329 in the discussion section as well, but that's less clear to me. In any case, this is far beyond my prerogative as a reviewer. I would merely advise that some of the material in this section move to a Results subsection and some remain here as discussion.

> We implemented the suggested reorganization.

l. 326-327 and Figure 10 legends: "The average difference between the two datasets is about 25% of the average value of Adusumilli et al. (2020) at 16 km.": I think the average of the difference is not the right metric to use here. Even if the average of the difference happened to be quite small (and I think that maybe should be the case here), that would not tell you very much about how different the datasets are. Instead, you should use the standard deviation of the difference here. That is a good measure of how different the data sets are from one another. Since that is about triple the mean melt rate (in either data set), it tells you there's a lot of disagreement.

> We agree, we removed the sentence and added in the text the following: "The standard deviation of the difference between the two datasets is 1.97 m.yr−1, which is about three time larger the mean melt rate of both datasets."

l. 329: "...by using the binning methods we give spatial freedom to the datasets and constrain them by their values." I think you need to state more clearly what "spatial freedom to the datasets" might mean. I also think it would be good to flesh out the discussion here even more about why you think it's important to get a good statistical representation of melt rates but why the details of the spatial pattern don't matter. More below...

> We developed with the following text:
>
> "Since calibration methods that minimise cell-to-cell differences between modelled and observed melt rates often fail to capture the overall distribution of observed values, and given the spatial inconsistencies among observational datasets, we prioritise reproducing the correct distribution of basal melt rates over minimising spatial mismatches. We consider that having a good statistical representation of the melt rates is potentially more important for the dynamic of the ice-sheet. For instance, the highest melt rate values are observed in the Amundsen sea area, where due to the retrograde slope the West Antarctic ice sheet is exposed to the marine ice sheet instability process (Weertman, 1974; Joughin et al., 2014), therefore capturing these high melt rates values are potentially important for future projections. Moreover, even if we might not have the right values at the right locations within ice shelves, we have seen in subsection 3.2 that the calibration method MAE of bins enable to have the values close to the distribution of the target in local areas. Prioritizing values over spatial correspondence is in agreement with Joughin et al. (2021) who argue ..."

l. 329-331: This is a great argument!

I think it would also be worth pointing out that the Joughin finding isn't a universally accepted result in the field, and instead some researchers (Gudmundsson 2013, doi:10.5194/tc-7-647-2013; Reese et al. 2017 https://doi.org/10.1038/s41558-017-0020-x for two) think that buttressing plays an important role and melt in more strongly buttressed areas has an outsized effect.

> We added explicitly some nuance in our statement as follow: "However, this is not universally accepted, other studies suggest that localized sub-ice-shelf melt can have a strong impact on the buttressing or that in more strongly buttressed areas sub-ice-shelf melt would have outsized effect (Gudmundsson, 2013; Reese et al., 2018b)"

Figure 10: In the caption, could you make clear that these results are for the 16 km GRISLI mesh? It wasn't clear to me until I looked at the supplementary material.

Also, could you help me understand why the difference between the means in panel a would be about 0.02 m/yr but the mean of the difference is 0.16 m/yr? My guess is that there are many cells where only one of the datasets has data, and that accounts for the difference but wanted to make sure. Otherwise, I would have expected the difference of the means to be the mean of the differences.

> This is right, we clarified the two above comments by adding the following lines in the caption: "Results on this Figure are shown for the mesh grid resolution of 16 km, more details are given for mesh grid resolution of 16 km and 40 km in supplementary materials section 13. The difference of the means (0.66 - 0.68 = - 0.02 m.yr−1) is different from the mean of the difference (0.16 m.yr−1) because in the mean of the difference only the grid cells with values in both datasets are taken into account."

Section 4.4: This is the first section within the discussion that I think fully belongs here. It is full of some really great points.

> Thank you.

l. 350-362: Be careful that you present these differences in a neutral tone. You can't know for sure whether the PICO or the quadratic nonlinear results are more accurate since we don't know the right answer for future response. The only room there is to criticize one parameterization in relation to another is based on the physics each includes or excludes.

> Thank you for noticing this, we value this to be done carefully. We therefore paid attention to this point when we rewrote the bullet points to answer to the following comments. Please let us know in case this has not be done properly enough according to you.

l. 350-351: "PICO includes overturning circulation under the ice shelves, which tends to reduce the basal melt rate. This freshwater negative feedback is not included in the QuadNL parameterisation." I think this is a misunderstanding of the quadratic parameterization. Like PICO, the quadratic parameterization has one term involving the thermal forcing that is meant to represent how warmer temperatures near the ice-ocean interface lead to higher melt rates. But the second term in the product involving the thermal forcing is meant to represent the stronger overturning that results from warmer conditions. This is essentially equivalent to the PICO term involving the overturning coefficient C. See, for example, Holland et al. (2008, doi:10.1175/2007JCLI1909.1), Jourdain et al. (2017, doi:10.1002/2016JC012509), Burgard et al. (2022, doi:10.5194/tc-2022-32). So I do not agree with the statement on this line and think it needs to be heavily revised or removed.

> We revised this point as follow:

"The overturning circulation under the ice shelves, which tends to reduce the basal melt rate, is computed differently in the parameterizations PICO and QuadNL. In PICO the overturning fluxes is computed with the overturning circulation coefficient C and the difference of densities (see equation 2). Whereas in the QuadNL it is in the product involving the thermal forcings which results in stronger overturning from warmer conditions (see equation (1) in Jourdain et al. (2020))."

l. 352-359: These are 3 excellent points!

Thank you.

l. 360-362: I really like the point about sensitivity to oceanic forcing in QuadNL. But I take issue with "therefore this QuadNL calibration could overestimate basal melt rates". I don't think this follows. The lack of refreezing certainly will mean that, if the method is calibrated to get the right mean melt, it will also underestimate melt maxima. But it doesn't follow that it will overestimate melt on the whole, this depends on the detail of the calibration approach. I would remove this phrase.

Thank you for pointing this out. We agree and remove this bullet point. We would like to bring attention to the new bullet point that discusses sensitivity in differently, it reads as follows:

"The sensitivity to ocean warming in our calibrated version of PICO lies below some previously reported ranges of Antarctic ice shelf sensitivity \citep{Levermann2020-esd, VanderLinden2023-tc}, and is more consistent with the PICO sensitivity range estimated by \citet{Lambert2024-tc}. In their study, the PICO sensitivity is lower than that obtained with other sub-shelf melt rate parameterisations, although it is not the lowest overall. By contrast, the QuadNL parametrisation lies on the higher end of sensitivity spectrum compared to other parametrisations \citep{Burgard2022-tc, Lambert2024-tc}. So far, neither PICO nor QuadNL sensitivities ranges, which also depend on their calibration, can be ruled-out as we do not know what the right sensitivity is."

This point is related to the new part 3.5 and new figure 8 that we show here (needed to response to the other reviewer's comments):

"Thanks to the ensembles with + 0 K and + 1 K ocean forcings from the previous subsection we can determine the sensitivity of PICO for all the ensemble members. We are here assuming a linear sensitivity and therefore compute it as the difference between the + 1 K experiment minus the + 0 K experiment. The results are shown in Figure 8 where we also differentiate the sensitivity in the 3 areas defined for the analysis shown in subsection 3.2. First of all, we see that in most cases the sensitivity of PICO increases when the value of either of the two parameters is increased. Second, we see that the sensitivity varies between areas. For instance, 1 K of warming with the calibration from PISM-PICO (squares

on the Figure 8) would lead to an increase of 1.5 m.yr−1.K−1 in Filchner-Ronne ice shelf, whereas it would be 8.4 m.yr−1.K−1 in the Bellingshausen and Amundsen seas ice shelves. We also see that the range of possible sensitivities (panel (d)) is about four times larger in the B.A. seas than in the FRIS. These results quantify how much the sensitivity of the basal melt rate would change, globally and regionally, by changing the values of the two PICO parameters. In all the cases, the best calibration with the MAE of bins (black hexagons) is in the low range of all the tested combinations of parameter values. These values are also lower than the range of Antarctic ice shelves sensitivity estimates from some previous studies (Levermann et al., 2020; van der Linden et al., 2023), but closer to the PICO sensitivity obtained by Reese et al. 2023; Lambert and Burgard (2024) when optimising parameters for present-day melting. The methodologies in the assessments of the sensitivities are however different in each study. Nonetheless, based on the results show in Figure 8 we can expect a low to moderate response of ice shelves in this calibrated version of PICO to future projections scenarios."

[Figure]

l. 363-364: I would think this would be fairly obvious to anybody doing ice sheet modeling. There is not necessarily any relationship between the area of floating ice (the ice shelves) and grounded ice, which is the only part that contributes to sea level. Even the relationship between the grounded area and the volume (or mass) above flotation can be complicated. But I have not generally seen the area of floating ice presented as a quantity of major dynamical significance for ice sheet models in the past. (But I will add that ice-sheet modeling is less my field.)

 Cf. following response.

l. 364-366: I'm afraid I don't quite follow this argument. It's not obvious to me why different time series of ice-shelf area would imply different pathways. Ice shelf area could be reduced along the same pathway just by the presence of more (or less) calving relative to grounding-line motion.

 Regarding the two above comments, we agree and removed the sentences. Instead, we included the following:

 "In addition, the results of the future simulations show that the ice shelves can have different behaviours depending on the calibration method and the choices of the values of the parameters. Thus, we here advocate for a calibration ..."

l. 368-369: "This methodology can potentially be applied to modules in other models that benefit from existing observational datasets." This is an excellent point! I think this work has exciting potential applications to other methods and models.

 Thank you.

l. 379: "...spatial contrast...": I find this phrase a little inexact. How about "range of special variability, if not the details of spatial patterns"?

 We corrected as suggested.

l. 390-391: This is an excellent point! The paleo record is one good way to constrain this work over longer times. Perhaps high fidelity coupled ice sheet-ocean simulations will provide another such method in the future.

 Thank you. We added the suggestion:
 "Thus, our analysis of future simulations is not enough to gain confidence in one specific projection, further work is needed to constrain the sensitivity of the Antarctic ice sheet using paleo records **and coupled ice sheet-ocean models**."

l. 398-399: I appreciate you providing this recommendation! I think many modelers will find it useful.

> Thank you.

l. 401-402: "But also, as the present study has the limitation of targeting a given basal melt rate for a given ocean temperature rather than the sensitivity of the basal melt rate to changes of ocean temperatures." I think this is a great point but it isn't clear how it follows from the previous sentence. How will more modelers testing the calibration method address this limitation?

> Those are two separate points. We clarify it by rephrasing this sentence:

> "Another possible improvement would be to target a sensitivity of the basal melt rate to ocean forcing changes rather than targeting a given basal melt rate for a given ocean forcing."

l. 404-406: "Alternatively, the low sensitivity of PICO in comparison to QuadNL could also be adjusted by developing a quadratic dependence to thermal forcing that would give a quadratic parameterisation that also accounts for overturning circulation under the ice shelves." Presumably, it will come as no surprise given my statements above that I do not agree with this statement. I believe the QuadNL parameterization does already take the overturning into account.

> We agree and removed this sentence.

**Typographical, Grammatical and Formatting Suggestions:**

l. 22: "Sub-surface melt of the ice shelves on the other hand…" It's not clear what "on the other hand" is contrasting here. Maybe something else like "in turn" is more appropriate?

> We simply removed "on the other hand"

l. 39: "…distribution of values from the observations as best as possible." The phrase "as best as possible" is often used informally but probably should be replaced with something like "to the extent possible" in scientific writing.

> Done as suggested.

l. 45: "…whether it matters to calibrate PICO at a smaller scale than Antarctic wide…" This took me a little time to understand. Maybe rephrase this for clarity as something like "…whether regional calibrations of PICO produce different results than those applied over all of Antarctica…"

> Done as suggested.

l. 65 and throughout the subsequent text and figures: "Sv.m3.kg−1": You consistently use periods to separate units. I imagine this is something the typesetter will handle but my experience with TC is that they use half-spaces (\, in latex) between symbols. I have

seen a multiplication dot (\cdot in latex) used occasionally in other journals but never a period. I would advise changing to a half space.

> We would like to adjust all the text and figures at the end, when the typesetter will request it.

l. 68: "under-burden pressure": This may just relate to our difference in point of view but I had not heard of this term used in this context before. I guess from an ice-sheet modeling perspective, this is the pressure of the ocean pushing up on the ice? From my ocean modeling perspective, this is the overburden pressure of the weight of the ice pressing down on the ocean. Maybe just call it the "pressure at the ice-ocean interface" or something similar to be agnostic to these two perspectives?

> The term was taken from the original PICO paper Reese et al. 2018, but we agree with the suggested formulation that is clearer.

l. 69: "pk = ρIce ∗ g ∗ zIceDraft": I would suggest using multiplication dots (\cdot in latex) here or just spaces but "∗" is not the right thing to use in scientific writing. For consistency with what you have on l. 74, ρIce should be ρI.

> Changed to \cdot, and the equation is now ρSeaWater so the second comment is not applicable anymore.

l. 75-77: "The coefficient a is the salinity coefficient of the freezing equation...pressure coefficient of the freezing equation and equals $7.77 \times 10^{-8}$ °C Pa$^{-1}$." This simplified equation is typically referred to as a linearization of the "equation of state for the freezing point of seawater". This might be better than "freezing equation." The coefficients are often called the liquidus slope, liquidus intercept and liquidus pressure coefficient, respectively (see, for example, Asay-Davis et al. (2016, doi:10.5194/gmd-9-2471-2016) – no need to cite this, just pointing to the terminology there).

> The text is corrected as follow: "The coefficients from linearisation of the equation of state for the freezing point of seawater are: a is the liquidus slope coefficient and equals $-0.0572$ °C.PSU$^{-1}$, b is the liquidus intercept coefficient and equals $0.0788$ °C, c is the liquidus pressure coefficient and equals $7.77 \times 10^{-8}$ °C.Pa$^{-1}$."

l. 83: "...we decide to make use of the drainage basins defined in (Mouginot and Rignot, 2017) and used in (Rignot et al., 2019)": These citations should be outside of parentheses (\citet instead of \citet in latex).

> This has been corrected.

l. 95: "All the four criteria are here not followed...": This would be significantly clearer as something like, "Here, we do not follow any of these criteria..."

> Adjusted as suggested.

l. 98: "…show in some areas refreezing…" should be "…show refreezing in some areas…"

Adjusted as suggested.

l. 102-103: "…we consider as grounding line any ice points that is surrounded by grounded ice and not grounded ice…": I found this a little hard to follow and I think it would be clearer as something along the lines of, "…we consider any ice points to be on the grounding line if it has some neighbors that are grounded and others that are floating…"

Adjusted as suggested.

l 108, 113, 210: "…enables to have…" and "…enables to limit…" should be something like "…enables us to have…" or "…enables one to have…" (less preferable). This particular passive form is not grammatically correct in English.

Adjusted with "enables us".

l. 124: "…30 years relaxation…" should be "…30-year relaxation…"

Adjusted as suggested.

l. 146 and 292; "We present first the three ones…" and "…the methods RMSE of bins and MAE of bins are the two ones..": It would be more appropriate for scientific writing to replace "three ones" and "two ones" with "three methods" and "two methods", respectively. (I know it sounds redundant, but there doesn't seem to be a good way to avoid that.)

Adjusted as suggested.

l. 151 and 153: "two Dimensions…" should be "two-dimensional"

Adjusted as suggested.

l. 159: "…we decide to proceed a binning on the dataset…": I'm not entirely sure what you are saying here but maybe, "…we decide to bin each dataset…"

l. 160 "method" should be "the method"

For the two comments above, we corrected the text as follow:

"To improve that, we decide **to bin each** datasets: the ensemble members as well as the target**. The aim of adding the binning** is to be able to force **the** method to pick ensemble members that fit better the target distribution, including the higher and lower tails of the distribution."

l. 161: Here and later in the text, I believe the standard is to use a capital F in "Figure" (and T in "Table") even within a sentence when referring to figures by number. I think you

do this later in the text in a few places, but mostly you use lowercase f, and I think you should use capital F throughout.

> This has been corrected in the whole paper.

l. 176-177: "For that…" isn't super clear or standard grammar. How about, "to accomplish this…"

> Adjusted as suggested.

l. 183: "…in line with…": This is super picky but I think this should just be "…using…" because "in line with" implies that the results are consistent with what you're about to refer to, not that they were produced using what you're about to refer to.

> We suggest the following rephrasing to be clear:
> "For these simulations, the PICO parameters values are consistent with the results of the analysis of the calibration ensemble defined in sub-section 2.5 and presented below in Figure 2."

l. 198, 200, 266, 271, 291 and elsewhere: "on panel" or "on figure" should always be "in panel" or "in figure".

> Corrected as suggested.

l. 207: "…will not be systematically in the same order of magnitude of values…" might be easier to follow as something like "…will not systematically be of the same order of magnitude…"

> Corrected as suggested.

Figure 3: I would put the panel letters before the text they refer to, not after: "…to each methods, (a) ADA of bins, (b) RMSE of bins and (c) MAE of bins…" This is more standard notation.

> Corrected as suggested.

l. 229: "…they even do not have values…" should be "…they do not even have values…"

> Corrected as suggested.

l. 235: "…the possibility to compensate…" should be "…the possibility that there is compensation…"

> Corrected as suggested.

l. 279: "result all" should be "all result"

> Sentence has been changed so not applicable anymore.

l. 283: "…discuss further about the ISMIP 2300 results obtained using PICO" should be "…discuss the ISMIP 2300 results obtained with PICO further."

> Corrected as suggested.

l. 290, 291, 313: "cumulating": Although "to cumulate" appears to be a valid English verb, as a native speaker I had never heard of it before. I would suggest using "accumulate," "aggregate," or "combine" instead.

> We adjusted all the cases with the suggested wording.

l. 294-296: This could be cleaned up a little as: "This consistency can matter to inter-compare: i) different parameterisations ; ii) when the same parameterisation is used in different ice sheet models ; iii) results at different resolutions." All 3 use "inter-compare" so it can be brought out front and "is" is missing.

> Corrected as suggested.

l. 306: An end parenthesis ")" is missing before the end of the sentence.

> Corrected.

l. 307: "...we present in figure 9 results..." should be "...in Figure 9, we present results..."

> The sentence is corrected to : "To remain concise, in Figure 9 we only present the results of the ranking for the MAE of bins method"

l. 318: "Hence, we could argue to choose a different target..." would be more grammatical as something like , "Hence, we could make an argument for choosing a different target..."

> Corrected as suggested.

l. 321: "They are overall similar.." should be, "Overall, they are similar..."

> Corrected as suggested.

l. 328: "...important to justify..." should be "...important in justifying..."

> Corrected as suggested.

l. 331: "...the resolution of the observational datasets..."

> Corrected as suggested.

l. 334-335: "...also consider to compute..." should be "...also consider computing..."

> Corrected as suggested.

l. 333-335: These are also great points of discussion.

> Thank you.

l. 345: I would suggest: "Elmer-ice with PICO even suggests a negative sea level contribution from Antarctica throughout the simulation."

> Corrected as suggested.

l. 347-348: "…with PICO as shown with the simulation with PISM." That's a lot of "withs". I'd suggest, "…with PICO as shown in the simulation with PISM."

> Corrected as suggested.

l. 348: "But here we want to provide…"

> Corrected as suggested.

l. 358: "…does not peak that high at grounding lines": again I take issue with the qualitative and subjective nature of this phrase. How about something like "…does not show significantly higher melt rates at grounding lines, as seen in satellite-derived fields"?

> Corrected as suggested.

l. 367: "fits best" should be "best fits"

> Corrected as suggested.

l. 377: "…forces to fit the target values also at the low and high extremes…" should be something like "…that also fit the low and high extremes in the target histogram…"

> Corrected as suggested.

l. 378: "systematic" should be "consistent"

> Corrected as suggested.

l. 380: "…matches well the entire distribution of…" could be something like "…does a good job at matching the entire distribution…"

> "good job" sounds a bit too familiar to us, maybe wrongly. We rephrase the sentence as follows: "By using these two methods that closely fit the entire distribution of the target for all Antarctic ice shelves combined, we also show that region-specific calibration is not necessary."

l. 388-389: "ice shelves extent" should be "ice-shelf extent" and "ice shelves changes" should be "ice-shelf changes"

> Corrected as suggested.

l. 394-395: "…could avoid modellers to use temperature corrections…": How about this? "…could avoid the need for modelers to use temperature corrections…"

> Corrected as suggested.

---

## Author Comment (AC2)

**Review for Menthon et al. : Comparison of calibration methods of a PICO basal ice shelf melt module implemented in the GRISLI v2.0 ice sheet model**

Submitted to Geoscientific Model Development
Reviewer: Clara Burgard

I do this review un-anonymously to clarify from which background and level of expertise the comments come from and because it makes the conversation during review more transparent on both sides.

> We would like to thank the reviewer, Clara Burgard, for taking the time to review the paper and provide detailed comments to improve the manuscript. We address all the points raised below, our responses to the comments are in blue.

**Summary**
Ocean-induced melt of Antarctic ice shelves and its representation in ice-sheet models is one of the main sources of uncertainty for ice-sheet projections. The parameterisations used to bridge the gap between ocean properties and the basal melt remain uncertain. The authors revisit this uncertainty and explore a new calibration approach to reduce it. To do so, they implement the PICO ice-shelf basal melt parameterisation into the ice-sheet model GRISLI and test different novel calibration approaches, resulting in a more robust parameter choice. They explore a variety of conditions that affect the calibration such as the metric, the resolution, the geographical specificities, and the forcing conditions.

The study is a timely contribution because the parameterisation of the melt at the base of Antarctic ice shelves remains one of the largest sources of uncertainty in projections of the future evolution of the Antarctic ice sheet and its contribution to sea-level rise. The approach explored by the authors is a useful addition to the community as it provides a set of parameters that seem to be more robust across conditions compared to previous sets. It could also be explored for other parameterisations in the future.

The manuscript is very clear and thorough. This thorough description is very well suited to inspire other researchers to explore the presented methods on other parameterisations and/or other ice-sheet models. Overall, I have a few larger comments as some aspects of the study appeared unclear to me and other more minor comments. I therefore suggest minor revisions.

**GENERAL COMMENTS**
I want to start by thanking the authors for making this manuscript very pleasant to as the manuscript and figures are very well done and clear.

**1 The results of this study are very encouraging. However, there is a strong limitation, which is not discussed much. In Figure 3, it becomes clear that using bins improves the melt rates for the "middle" bins but, in all cases, the spread between members remains high for the low and high-end bins. In particular, for the lowest melt, an anomaly of 0.5 m per yr could be quite high compared to the actual melt. For the highest melt, the order of magnitude is maybe lower compared to the actual melt. However, it is important to check this further because the points with the highest melt are also the points that influence the future of the ice sheet the most. Would it be possible to complete this by a metric that**

looks at the percentage formed by this anomaly compared to the melt value? Just to have a better idea of what this means exactly.

We agree, therefore we completed the figure with 3 panels that look specifically at the lowest bin (b), the highest bin (d) and the sum of of the absolute error of the 10 bins (f). All the results are expressed in percentages. We observe different responses between the lowest and highest bins and the new metric is in really good agreement with the results of calibration with the MAE of bins. We therefore added the following text. Also, this analysis has been done with the Adusumilli et al. 2020 datasets (shown in the main text) as well as the Paolo et al. 2023 datasets (added to the supplementary materials), both datasets leads to the same conclusions.

"The largest discrepancies between the binned values of modelled melt rates and those from observational datasets occur at the extremes, that are the bins with the lowest 10% and highest 10% of the values from the distribution. We therefore analysis further these two bins for all the PICO configurations in panels (b) and (d) of Figure 3, respectively. These panels reveal distinct sensitivities to the two PICO parameters. For example, at fixed values of C, increasing $\gamma*T$ consistently raises the bin values in both the lowest (panel (b)) and highest (panel (d)) deciles. This implies a reduction in error when the model underestimates melt (negative error, in blue), or an amplification of error when it overestimates melt (positive error, in red). In contrast, at fixed $\gamma*T$, varying C can produce divergent effects between the lowest and highest bins. For instance, at $\gamma*T = 2.0 \times 10^{-5}$ m,s$^{-1}$, increasing C leads to a decrease of bin error values in the lower 10% bin (b) and an increase in bin error values in the upper 10% bin (d). Finally, by computing the sum of the absolute values of the errors for all 10 bins we can find the combinations of PICO parameters that minimize this error the most. The panel (f) of Figure 3 shows the results with superposed the best members according to the MAE of bins methods shown on panel (e). We find that the two metrics, MAE of bins and sum of absolute errors of the bins, leads to a similar selection of the best ensemble members."

[Figure]

**2 I am not completely convinced by section 4.2. I do not understand how the authors can calibrate on different input temperatures (e.g. 1K warmer) but same target melt and make conclusions from this. If the parameters do not change much, does that not mean that they are not sensitive enough to temperature changes? Ideally, you would expect higher melt for higher temperatures if you use the same set of parameters, no? This makes me unsure about the whole approach. Does that mean that the parameterisation cannot react to changes in forcing and that the parameters are too strongly set? This would not be useful for projections and is what could be interpreted from the low sealevel rise contribution in Fig. 7c. To reassure the reader, I recommend that the authors clarify the implications of this result, discuss them more in detail or reformulate to avoid misunderstanding.**

We agree with this concern, to answer and quantify it we decided to add a new subsection (with a new text and a new figure) about "What is the sensitivity of PICO?". In the new figure we quantify the sensitivity of PICO thanks to the ensembles with + 0 K and + 1 K forcings, and we separate the sensitivity of the 3 areas of interests used earlier.

The following text is added at the end of the subsection 3.4
"Since the best parameters do not vary much (about - 0.5 for the $\gamma * T$ and - 0.05 for C) between the + 0 K and the + 1 K forcings, we analyse in the next subsection what is the sensitivity of PICO to understand better what would be the response of PICO to warmer than present-day conditions."

And the following text and the Figure with sensitivities values is the subsection 3.5:

"Thanks to the ensembles with + 0 K and + 1 K ocean forcings from the previous subsection we can determine the sensitivity of PICO for all the ensemble members. We are here assuming a linear sensitivity and therefore compute it as the difference between the + 1 K experiment minus the + 0 K experiment. The results are shown in Figure 8 where we also differentiate the sensitivity in the 3 areas defined for the analysis shown in subsection 3.2. First of all, we see that in most cases the sensitivity of PICO increases when the value of either of the two parameters is increased. Second, we see that the sensitivity varies between areas. For instance, 1 K of warming with the calibration from PISM-PICO (squares on the Figure 8) would lead to an increase of 1.5 m.yr−1.K−1 in Filchner-Ronne ice shelf, whereas it would be 8.4 m.yr−1.K−1 in the Bellingshausen and Amundsen seas ice shelves. We also see that the range of possible sensitivities (panel (d)) is about four times larger in the B.A. seas than in the FRIS. These results quantify how much the sensitivity of the basal melt rate would change, globally and regionally, by changing the values of the two PICO parameters. In all the cases, the best calibration with the MAE of bins (black hexagons) is in the low range of all the tested combinations of parameter values. These values are also lower than the range of Antarctic ice shelves sensitivity estimates from some previous studies (Levermann et al., 2020; van der Linden et al., 2023), but closer to the PICO sensitivity obtained by Reese et al. 2023; Lambert and Burgard (2024) when optimising parameters for present-day melting. The methodologies in the assessments of the sensitivities are however different in each study. Nonetheless, based on the results show in Figure 8 we can expect a low to moderate response of ice shelves in this calibrated version of PICO to future projections scenarios."

[Figure]

**DETAILED CONTENT COMMENTS**

L3 : It is not only about limited computational resources but also about a lack of observational data. I suggest completing "limited observational data and computational resources".

> Now the sentence reads as follow: "As a result of limited scientific understanding of these ice-ocean interactions, poor observational data, existing biases in earth system models, as well as technical challenges related to computational resources and implementation, these interactions are parametrized rather than explicitly resolved in most ice sheet models."

L31: Not sure if the formulations only differ in the complexity of the "melt physics" themselves. In most cases, assumptions differ on the simplification of the ocean circulation in the cavity. Can the authors reformulate to clarify?

> Now the sentence reads as follow: "Over the last decade, several basal melt parameterisations have been developed and implemented in ice sheet models with different complexities in the simplification of the ocean circulation beneath the ice-shelves and its physical interaction with the ice."

L39-40: This is great! Calibrations that do not need regional corrections are what are currently needed!

> Thank you.

L104-108: This paragraph is unclear in regard to the authors' own contribution vs what has been done in PICO-PISM before. I suggest reformulating it to clarify.

> Clarification were made as follow, with the new parts in bold: **"The grounding line can be defined in different ways and therefore can lead to different PICO boxes geometries. Here, in GRISLI-PICO,** we consider as grounding line any ice points that is surrounded by grounded ice and not grounded ice, and as ice front any ice point that is floating and adjacent to ocean. **Whereas, in** PISM-PICO (Reese et al., 2018) they did not include the grounding line of ice rises and also excluded holes in ice shelves as ice shelves front when identifying PICO boxes. The grounding lines of ice rises are defined as not being directly connected to the main grounded part of the ice sheet which is identified by the size of the connected grounded region. Thus, **in PISM-PICO i**t is possible to have ice shelves without grounding line connected to the main ice sheet, where PICO cannot define a box geometry. In these places, the parametrisation of Beckmann and Goosse (2002) **was used in PISM** to have a rough estimate of the basal melt rates."

L114: It is unclear where the value for the combined factor comes from. As far as I know $K\_T$ is a parameter that needs calibration. So how has it been calibrated in this case? I suggest the authors add the source or briefly explain the calibration.

> No additional calibration have been done for this $K\_T$, therefore we clarified the sources:
> The combined factor $K_T \rho_w C_w \rho_i L_f$ equals to 0.224 m.yr$^{-1}$.$^{\circ}$C$^{-2}$ as we keep the same $K_T$ value of 15.77 m.yr$^{-1}$.C$^{-1}$ from DeConto and Pollard (2016) and Pollard and Deconto (2012).

L122-124: It remains unclear to me why the authors use the coupled GRISLI-PICO setup if the geometry is kept constant anyway. Could the authors clarify why they do not use a standalone version of PICO then?

It could have been done in a stand-alone PICO indeed. This choice has been made because the goal is to use PICO coupled with GRISLI as quickly as possible. Doing the coupling PICO with GRISLI leads to some changes (reading variables differently etc.), therefore doing the stand-alone and the coupled would have lead to more work for very similar results. We added the following in the text:
"Doing the calibration of PICO in a coupled GRISLI-PICO with fixed geometry enables to facilitate the transition between the PICO calibration and the GRISLI-PICO transient experiments (see section \ref{m-future}) without impacting the results."

Table 1: Very useful table!

Thank you.

Table 1: Maybe I did not think this completely through but aren't ADA and ADA of bins the same metric. Taking the mean of a mean should, I think, result in the same as taking the mean of the whole ensemble directly. This would also explain why the results are so similar between the two. Can the authors comment on that?

We agree with your point. The very small differences observed between the two methods is due to the sampling of the bins, in other words how many grid cells values are per bins, and despite this the weight given to each bin will remain the same.
For instance:
- (1+2+3+4+5+6+7+8+9+10)/10=5.5
- Bin 1: (1+2+3)/3=2
- Bin 2: (4+5+6)/3=5
- Bin 3: (7+8+9+10)/4=8.5
- Average of the bins=5.16

We therefore clarified this with the following text in the paper:

"This method do not leads to exactly the same results as the ADA without binning because some bins contain slightly different numbers of grid cells than others, meaning that they cover slightly different areas but still get exactly the same weighting in the binning approach. As we intend to compare methodologies, we consider it relevant to also test this method, even if it gives results very close to ADA without binning, to quantify how much they differ."

L218-219: I am not 100% sure that a narrow set of parameters is a guarantee for "better". I agree it is useful for modellers but I would be careful with such kind of statements. If a large range of parameters are possible, this could also be linked to the formulation of the parameterisation. Still, in L294-296, the authors explain the advantages of having a narrow set of parameters. Maybe this could already be mentioned here?

We argue that a narrow selection of best parameter sets proves that the results will be less dependent on the sampling choices for the ensemble, therefore the calibration methodology is better. However, we agree that it does not necessarily means that the

values of the parameter defined by the methodology will be the right ones for the rights reasons, the parametrisation remains an (imperfect) simplification of the physical processes. On the other hand, if a parametrization works equally good or bad for any parameter value, then the parametrization isn't useful, so in that sense a parametrization with a narrow range of parameter values is better.

L241: It is unclear to me what is the difference between the distribution curve and the magnitude of the spatial patterns. Is one the shape of the curve and the other the actual number?

No, the distribution curve is what is shown on figure 2 panels a, b, c g, h and i (the compilation of all the values regardless to their geographical coordinates); whereas the magnitude of the spatial patterns is shown on figure 4 (the values at their geographical coordinates). We therefore help the reader by referencing to the figures:

"[…] i) better able to match the distribution curve from the target (Figure 2, panels (a), (b), (c), (g), (h), (i)), ii) systematically give best values in the same small range of values (Figure 2, panels (d), (e), (f), (j), (k), (l)), and iii) the magnitude of the spatial patterns is similar to the target (Figure 4)."

Figure 8: Just a remark: The 2D results are interesting and give the feeling that C does not really play a role in the calibration. Out of curiosity, have the authors thought about what this could mean?

Good question, to which we do not have a definitive answer. One idea could be that for higher gamma, PICO extracts heat from the available ocean forcing quite efficiently, and the limitation on melting is how much heat is supplied (which is constrained by C). For lower gamma values, less heat is extracted from the ocean reservoir, and hence the overturning does play less of a role.

L330: Agreed that Joughin et al. (2021) is a good study to refer to here but it should not be forgotten that enough other studies (e.g. Reese et al. 2018) show that localised melt has a strong effect on buttressing. I suggest that the authors reformulate a little more carefully.

We added explicitly some nuance in our statement as follow: "However, other studies suggest that localized sub-ice-shelf melt can have a strong impact on the buttressing or that in more strongly buttressed areas sub-ice-shelf melt would have outsized effect (Gudmundsson, 2013; Reese et al., 2018b)"

L350: This is not completely true. The quadratic term of the quadratic parameterisation is there to mimic the effect of the overturning circulation in a very simple way.

We revised this point as follow:

"The overturning circulation under the ice shelves, which tends to reduce the basal melt rate, is computed differently in the parameterizations PICO and QuadNL. In PICO the overturning fluxes is computed with the overturning circulation coefficient C and the difference of densities (see equation 2). Whereas in the QuadNL it is in the

product involving the thermal forcings which results in stronger overturning from warmer conditions (see equation (1) in Jourdain et al. (2020))."

L360-362: Yes, it can be clearly seen in the sensitivities to warmer forcing in Burgard et al. (2023) and Lambert and Burgard (in press) that the quadratic parameterisation is an outlier towards high melt sensitivity. However, as we do not know what is the "right" sensitivity, this is not enough to say that one is better than another.

We completed the argument, the whole point reads as follow:

"The QuadNL calibration from Jourdain et al. (2020) implemented in the analysed simulations does not match the refreezing part of the observations whereas PICO does (see figure 6 panel (b)), therefore this QuadNL calibration could overestimate its sensitivity to oceanic forcings, which is in agreement with recent studies (Burgard et al., 2022; Lambert and Burgard, 2024). However, even if the QuadNL parametrization is on the high-end in terms of sensitivity compared to other parametrizations, neither PICO nor QuadNL sensitivities (which also depend on their calibration) can be rule-out as we do not know what the right sensitivity is."

L363-366: This is not very clear. I suggest that the authors reformulate to clarify.

**DETAILED WRITING COMMENTS**
L20: Replace "warmth" by "heat"
    Done as suggested
L22: "on the other hand" does not really work in this sentence. I suggest leaving it out.
    Done as suggested
L34: Leave out "However", it is confusing.
    Done as suggested
L41-49: This could be shortened.
    It has been shortened as follow:
    "The article presents first the methodology in section 2, then the results in details in section 3 including: calibrations methods comparison, sensitivity estimates and future projections. The section 4 discusses the limitations and possible improvements for next studies. Finally, section 5 concludes and gives some perspectives."
L54: Replace "are dependent" by "depend"
    Done as suggested
L79 and 84: For the results, I suggest to stay consistent with present instead of past tense: "made" => "make", "combined" => "combine"
    Done as suggested
L83: Correct the citation format (\citet{} instead of \citep{})
    Done as suggested
L98: Rephrase to "refreezing in some areas"
    Done as suggested
L142: Replace "but also" by "and"
    Done as suggested
L148: To improve reproducibility, I suggest that the authors add the information if the results are in m ice per year or in m w.e. per year.
    We added: "(m of ice equivalent per year)"
L158: I suggest that the authors reformulate, the formulation is very unclear.

> It has been rephrased as : "By applying the three first ranking methods, the ranking metrics do not enable to pick systematically the ensemble members with the best fit to the distribution of values of the observational dataset."

Figure 1 caption, last sentence: Replace "is" by "are"

> Done as suggested

L190: Can the authors clarify if they are writing about the sensitivities of PICO and QuadNL to ocean warming or to something else?

> This rephrased as follow for clarity: "the difference of sensitivity between PICO and QuadNL"

L209 and later: "side-by-side" sounds awkward. I suggest reformulating, maybe with something like "close" or "similar".

> We disagree, close or similar would mean that the points are close but not necessarily "touching" each other, which is what we are trying to say here.

L220: Replace "explain" by "explained"

> Done as suggested

L224-225: I suggest reformulating as this is not a complete sentence.

> We suggest the following:
> "Whereas, the RMSE of bins and the MAE of bins have systematically smaller anomaly values and do not allow for compensating effect."

L232: "led" => "lead"

> Done as suggested

L235: remove "is"

> Done as suggested

L236: missing "methods" in front of "without"

> Indeed, done as suggested

L269: "Elemer" => "Elmer"

> Corrected as suggested

Figure 6 and later: Would it be possible to replace CalibXX by an indication of the metric it was calibrated with? That would clarify the legend.

> We understand this suggestion and we tried it. However, it makes the text a lot less clear (need to mention two values instead of one every time we want to refer to one simulation). Also, we believe it is important that the reader understand where the calibration is located in the parameter space and therefore it might be good to force the reader to come back to Figure 10 (a).

L284: I suggest reformulating "we further discuss the ISMIP 2300 …"

> Changed to: "Lastly, we discuss the ISMIP 2300 results obtained with PICO further."

L359: "rates" => "rate"

> The sentence has been changed to: "PICO tends to have more smoothed out melt rates and does not show significantly higher melt rates at grounding lines, as seen in satellite-derived fields."

L401-402: This is not a complete sentence.

> We believe it is a complete sentence.

Supplementary material: There is A LOT of material and the captions are sometimes very short. I wonder if it would be possible to reduce the amount of figures or add one sentence explaining the core of the figure or set of figures when appropriate?

> We strongly reduced the number of figures by removing all the figures showing results for the methods: 2D RMSE, ADA of bins and RMSE of bins. We argue that they show similar results to 2D MAE, ADA and MAE of bins (respectively), we therefore kept only the three later onces.

Moreover, we added an introduction to the supplementary to justify this choice and help the reader through the supplementary materials.

However, please note that in order to reply to the review thouroughly, two figures have been added to the supplementary materials: Figure S1 showing the PICO boxes and the drainage basins division, and Figure S29 showing the analysis of the bins but with target the datasets from Paolo et al. 2023 instead of Adusumilli et al. 2020.

References
- Reese et al. 2018 : https://doi.org/10.1038/s41558-017-0020-x
- Burgard et al. 2023: https://doi.org/10.1029/2023MS003829
- Lambert and Burgard, in press: https://doi.org/10.5194/egusphere-2024-2358

---

## Referee Report (RR1)

Second round of review for Menthon et al.: **Comparison of calibration methods of a PICO basal ice shelf melt module implemented in the GRISLI v2.0 ice sheet model**

Submitted to *Geoscientific Model Development*

I thank the authors for answering my questions and taking my comments into account. The manuscript has improved since the last version. The unclear points have been clarified and the restructuring is appropriate. I have a few minor comments left but, all in all, I think the manuscript is ready for publication.

**L35-36:** This sentence is redundant. I would suggest something along the lines of :"The parameterisations rely on the definition of their parameters." But even this sounds awkward. Maybe leave it out altogether?

**L106-108:** This is not a sentence. If a sentence starts with "whereas", the second part of the sentence should be in contradiction with the first part. I don't think it works they way the authors use it here. But this can be left to the proofreading maybe?

**L167-168:** This is not a complete sentence either. Or it is formulated in a very convoluted way that got me lost.

**L171:** The word "adding" could be removed here.

**L242:** "analysis" => "analyse"

**Figure 3 caption:** "bins values" seems imprecise. Can the authors reformulate in a clearer way?

**L253-260:** Using past tense here for the verb "leading" in several instances makes the reading a little difficult.

**L335-348:** The answer to the question of the title is hidden in a large text about observational differences. I would suggest either merging this section with another one or finishing it with the main conclusion of the paragraph (what is now at L340-344). At the moment, it reads mainly like a description of the differences between the two observational datasets.

**L388-390:** There is a mix-up of sentences here.

**L448:** The melt sensitivities to warming were explored in Burgard et al. 2023 and not in Burgard et al. 2022.

**L459:** I suggest adding "simulations of" before "future dynamics of the ice sheet"

**L488-490:** This sentence is tedious to read. Can the authors reformulate?

**References**

- Burgard et al. 2023: https://doi.org/10.1029/2023MS003829

---

## Author Response (AR2)

Second round of review for Menthon et al.: Comparison of calibration methods of a PICO basal ice shelf melt module implemented in the GRISLI v2.0 ice sheet model Submitted to Geoscientific Model Development

We, the authors of the paper, thank the two reviewers for their time and comments to finalize this paper. We respond to each comment in blue in the text below.

**From reviewer Clara Burgard:**

I thank the authors for answering my questions and taking my comments into account. The manuscript has improved since the last version. The unclear points have been clarified and the restructuring is appropriate. I have a few minor comments left but, all in all, I think the manuscript is ready for publication.

L35-36: This sentence is redundant. I would suggest something along the lines of :"The parameterisations rely on the definition of their parameters." But even this sounds awkward. Maybe leave it out altogether?

Indeed, we removed it as suggested.

L106-108: This is not a sentence. If a sentence starts with "whereas", the second part of the sentence should be in contradiction with the first part. I don't think it works they way the authors use it here. But this can be left to the proofreading maybe?

We suggest changing it to the following:

"In contrast, in PISM-PICO Reese et al. (2018a) did not include the grounding line of ice rises ..."

L167-168: This is not a complete sentence either. Or it is formulated in a very convoluted way that got me lost.

We agree and rephrased the sentence as: Since no squaring is used in the error computation of the MAE, the MAE is less sensitive to outliers than the RMSE.

L171: The word "adding" could be removed here.

Done as suggested.

L242: "analysis" => "analyse"

Done as suggested.

Figure 3 caption: "bins values" seems imprecise. Can the authors reformulate in a clearer way?

We rephrased as follow: "Anomaly of the values of the bins for the five best members..."

L253-260: Using past tense here for the verb "leading" in several instances makes the reading a little difficult.

We changed the tense to present and the verbs:

We see that four methods (ADA, ADA of bins, 2D RMSE, and 2D MAE) **results in a** spatial distribution with little contrast between higher and lower values, they do not

even have values more negative than -1 m.yr\$^{-1}\$ in blue (Figure \ref{fig\_MAPS-global-R16}) panels (a) to (c)). It could be because this selection **gives** low \$\gamma\_{\mathrm{T}}^{\*}\$ values, \$0.1\times10^{-5}\$ m.s\$^{-1}\$ and \$0.25\times  $10^{-5}$ \$ m.s\$^{-1}\$. Whereas, the best single member following the RMSE of bins or the MAE of bins have a lot more contrast (Figure \ref{fig\_MAPS-global-R16}) panels (d) and (e)), which corresponds better to what is seen in the observations (Figure \ref{fig\_MAPS-global-R16}) panel (f)). These two methods **results in** higher \$\gamma\_{\mathrm{T}}^{\*}\$ values: \$1.5\times10^{-5}\$ m.s\$^{-1}\$ and \$2.0\times 10^{-5}\$ m.s\$^{-1}\$.

L335-348: The answer to the question of the title is hidden in a large text about observational differences. I would suggest either merging this section with another one or finishing it with the main conclusion of the paragraph (what is now at L340-344). At the moment, it reads mainly like a description of the differences between the two observational datasets.

We added the following sentence to conclude clearly the paragraph:

"Hence, despite the important spatial differences, we conclude that the calibration methods are not significantly sensitive to the choice of the target dataset as they both give a similar selection of best ensemble members for each method."

L388-390: There is a mix-up of sentences here.

Indeed, thank you. We corrected and two mix-up sentences are the following sentence:

"Prioritizing values over spatial correspondence within an ice shelf is in agreement with \cite{Joughin2021-sa} who argue that the ocean-induced melt volume, regardless of the spatial distribution, directly paces the ice loss."

L448: The melt sensitivities to warming were explored in Burgard et al. 2023 and not in Burgard et al. 2022.

Done as suggested.

L459: I suggest adding "simulations of" before "future dynamics of the ice sheet" Done as suggested.

L488-490: This sentence is tedious to read. Can the authors reformulate? We reformulated as follows:

Reese et al. 2023 calibrated PICO to a sensitivity to temperature, but it required the use of temperature corrections. We suggest that using the MAE of bins calibration method could enable calibrating to sensitivity without additional temperature corrections.

**References**

- Burgard et al. 2023: https://doi.org/10.1029/2023MS003829

**From reviewer Xylar Asay-Davis:**

The authors have done an excellent job of addressing my recommendations. The manuscript looks to be in good shape to publish!

I have one small change that I would recommend but I am fine if the authors don't choose to include it.

l. 150 of the "diff" document: I would change "30-years relaxation" to "30-year relaxation". I had asked for an explanation of that relaxation and the authors kindly provided it in the response to my review but no further explanation is provided in the text, and I think some of the clarification I was given would also be of interest to readers.

Done as suggested. The text reads now as follows:

"The geometry of the ice sheet and the ice shelves is kept fixed to remove the influence of ice shelves geometry changes on the computed basal melt rate. The fixed geometry corresponds to Bedmap2 \citep{Fretwell2013-tc} with a 30-year relaxation with GRISLI. This relaxation is needed because with the basal drag coefficient inversion methodology used for ISMIP6, we compute the ice sheet internal thermal equilibrium with a long (60 kyr) experiment with fixed observed geometry. Thus, to avoid any artificial drift when releasing this constraint we run a 30 years relaxation experiment with the same boundary conditions as for the control experiment from ISMIP6 \citep{Seroussi2024-ef}."